# Reward Poisoning on Federated Reinforcement Learning

**Evelyn Ma**                                                            *pingm@illinois.edu*
*Department of Industrial and Systems Engineering*
*University of Illinois Urbana-Champaign*

**S. Rasoul Etesami**                                                    *etesami1@illinois.edu*
*Department of Industrial and Systems Engineering, Coordinated Science Lab*
*University of Illinois Urbana-Champaign*

**Praneet Rathi**                                                        *prathi3@illinois.edu*
*Department of Computer Science*
*University of Illinois Urbana-Champaign*

## Abstract

Federated learning (FL) has become a popular tool for solving traditional Reinforcement Learning (RL) tasks, particularly when individual agents need to collaborate due to low sample efficiency but are concerned about data privacy. The multi-agent structure addresses the major concern of data-hungry in traditional RL, while the federated mechanism protects the data privacy of individual agents. Despite the advantage FL brings to RL, Federated Reinforcement Learning (FRL) is inherently susceptible to poisoning, as both FL and RL are vulnerable to such training-time attacks; however, the vulnerability of FRL has not been well-studied before. In this work, we propose a general framework to characterize FRL poisoning as an optimization problem and design a poisoning protocol that can be applied to policy-based FRL. Our framework is versatile, catering to FRL scenarios employing both policy-gradient local RL and actor-critic local RL. In the context of actor-critic configurations, we conduct training for a pair of critics, one private and one public, aimed at maximizing the potency of poisoning. We provably show that our method can strictly hurt the global objective. We verify the effectiveness of our poisoning approach through comprehensive experiments, supported by mainstream RL algorithms, across various RL OpenAI Gym environments covering a wide range of difficulty levels. Within these experiments, we assess our proposed attack by comparing it to various baselines, including standard, poisoned, and robust FRL methods. The results demonstrate the power of the proposed protocol in effectively poisoning FRL systems – It consistently diminishes performance across diverse environments, proving to be more effective than baseline methods. Our work provides new insights into the training-time vulnerability of FL in RL and poses new challenges for designing secure FRL algorithms.

## 1  Introduction

*Reinforcement Learning* (RL) has gained popularity in recent years as a paradigm for solving complex sequential decision-making problems and has been applied to a wide range of real-world problems, including game playing (Silver et al., 2016; Vinyals et al., 2019), autonomous driving (Yurtsever et al., 2020), network security (Xiao et al., 2018), design of materials (Govindarajan et al., 2024; Zhou et al., 2019; Ghugare et al., 2023), circuit design (Roy et al., 2021), and optimization (Chen et al., 2022). In RL, the agent's goal is to learn an optimal policy that maximizes the long-term cumulative rewards, which is done by repeatedly interacting with a stochastic environment, taking actions, and receiving feedback in the form of rewards. However, despite the impressive performance of RL algorithms, they are notoriously known to be data-hungry, often suffering from poor sample efficiency (Dulac-Arnold et al., 2021; Schwarzer et al., 2020). One traditional solution to this challenge is *Parallel RL* Kretchmar (2002), which adopts multiple parallel RL agents that

sample data from the environment and share it with a central server, as seen in practical implementations such as game-playing (Mnih et al., 2016; Berner et al., 2019). However, transferring raw data may not be feasible: on the one hand, it can cause significant communication costs in applications, such as the Internet of Things (IoT) (Wang et al., 2020); On the other hand, it is not suitable for privacy-sensitive industries, such as clinical decision support (Liu et al., 2020).

*Federated Reinforcement Learning* (FRL) has been proposed in order to address the limitations of traditional parallel RL, inspired by the recent success of *Federated Learning* (FL). FRL allows multiple agents to solve the same RL task collaboratively without sharing their sensitive raw data, thus addressing the drawbacks of heavy overhead and privacy violation in traditional Parallel RL. On the communication side, the communication efficiency of FRL is improved by the ability of FL to perform multiple local model updates during each communication round. On the privacy side, FRL enables data privacy protection by only communicating model updates, but not raw data, to a central server. With advantages of addressing communication efficiency and protecting privacy, FRL is practically appealing to a wide range of applications, including IoT (Wang et al., 2020), autonomous driving (Liang et al., 2023), robotics (Liu et al., 2019), etc. The success of FRL applications has spurred theoretical and methodological advancements. For instance, Mnih et al. (2016); Min et al. (2023b;a) investigate FRL under asynchronous communication, while Xie & Song (2023b); Zhu & Gong (2023); Xie & Song (2023a); Mai et al. (2023); Fan et al. (2023) delve into FRL with heterogeneity in local environments. Khodadadian et al. (2022); Zheng et al. (2023) provides insights into linear speed-up convergence rates, Woo et al. (2023) proposes an algorithm with sample complexity linearly scaling with the number of agents, and Zhao et al. (2023); Liu et al. (2023) analyze FRL with personalized local objectives.

Poisoning in FRL is practical and harmful. The inherent nature of FL and RL amplifies the susceptibility to poisoning (training-time attacks) when combined into FRL. On the RL side, individual RL agents are designed to dynamically learn from raw feedback from the environment, making the learning system vulnerable to data corruption during training time (Zhang et al., 2020; Banihashem et al., 2022). As an example, a chatbot could be misled by a small group of Twitter users, resulting in the generation of misogynistic and racist remarks (Neff, 2016). Similarly, recommendation systems can be manipulated by a minimal number of fake clicks, reviews, or comments, leading to products being inaccurately ranked higher. Besides, RL in material design (Govindarajan et al., 2024; Zhou et al., 2019; Ghugare et al., 2023) is susceptible to poisoned material data, and RL in circuit design (Roy et al., 2021) can be compromised by misleading prefix graphs and actions. On the FL side, the impact of poisoning is exacerbated in FL frameworks compared to single-agent RL. The lack of transparency in local training within FL naturally exposes the FRL system to adversarial attacks by malicious agents (Fang et al., 2020; Bagdasaryan et al., 2020; Bhagoji et al., 2019; So et al., 2020).

Although practical and harmful, poisoning in FRL has yet to be explored. In this work, our goal is to *systematically study the vulnerability of FRL systems when facing adversarial attacks that attempt to mislead the trained policy.*

In the remainder of this introduction, we discuss the challenges of directly applying prior RL poisoning methods to FRL in Section 1.1, review related work on poisoning in Section 1.2, and summarize our contributions in Section 1.3.

## 1.1 Challenges of applying prior RL poisoning to FRL

Applying existing RL poisoning techniques to FRL presents several challenges. Many prior methods, such as those by Zhang et al. (2020) and (Rakhsha et al., 2020), rely on unrealistic assumptions, including the attacker having complete knowledge of the MDP environment, which is often impractical. Additionally, some approaches, such as TrojDRL (Panagiota et al., 2020), target RL agents' actions rather than rewarding schemes, making them incompatible with our framework. Furthermore, the effectiveness of certain mechanisms, such as VA2C-P(Sun et al., 2020), diminishes in federated settings due to small local training steps, which lead to frequent reinitialization of the adversarial critic and impaired learning of the poisoned actor's value function. Nevertheless, we have included the federated version of VA2C-P as a baseline, and our experimental results highlight the advantages of our approach compared to this federated extension of RL poisoning.

## 1.2 Related Work on Poisoning

Here, we provide a comprehensive literature comparison regarding *poisoning* in various contexts of FL, RL, Multi-Agent RL, Distributed RL, and FRL.

*Poisoning in FL* has been analyzed in different settings. Selected representative studies (Tolpegin et al., 2020; Bhagoji et al., 2019; Fang et al., 2020; Jodayree et al., 2023b;a) strategically poison FL by either compromising the integrity of the training dataset or manipulating submitted local model parameters. However, these studies are under the context of Supervised Learning, which is substantially different from RL in the sense of the availability of future data, the knowledge of the dynamics of the environment, etc (Sun et al., 2020). Thus, the existing works on FL poisoning are not directly applicable to FRL poisoning. In contrast, our work focuses on the poisoning designed specifically for FRL, taking into account the unique characteristics of RL.

*Poisoning in RL*, referring to committing attacks during the training process (Zhang et al., 2020), can be categorized into two types: weak attacks that only poison data (e.g., rewards and states) and strong attacks that can manipulate actions in addition to data (Panagiota et al., 2020). In this study, we focus on weak attacks, also known as environmental poisoning, as they allow easier access for the attacker and, therefore, are more likely to occur in real-world scenarios. Rakhsha et al. (2020) formulated optimization frameworks to characterize RL poisoning, but they have limitations, such as requiring knowledge of the underlying MDP and focusing on targeted poisoning. Our proposed framework, however, considers both targeted and untargeted poisoning under the realistic assumption that the attacker does not have access to the MDP dynamics. Sun et al. (2020) designs a vulnerability-aware poisoning method for RL. Their algorithm focuses on manipulating the actor model, which cannot be directly applied to FRL as the critic model is also communicated among agents and the server. In contrast, our proposed poisoning mechanism is specifically designed for FRL and focuses on manipulating the critic model by training a pair of public and private critics.

*Poisoning in Multi-Agent Reinforcement Learning (MARL)* is typically different from poisoning in FRL. Existing literature studying the robustness or poisoning of MARL more or less violates the security code in FL. In MARL, multiple agents' actions jointly determine the next state of the environment and the corresponding rewards for each agent, thus exposing data and environment between agents (Wu et al., 2023; Liu & Lai, 2023; Mohammadi et al., 2023; Li et al., 2023). On the contrary, our work strictly adheres to the privacy-oriented settings in FRL, letting the agents independently engage in exploration and data collection solely within their local environments, thus ensuring that no data or environment is exposed to other agents.

*Poisoning in Distributed RL*, similar to poisoning in MARL, can violate the privacy code within the FRL context. Previous studies on poisoning or robustness in Distributed RL can expose data to the central server. For instance, Chen et al. (2023) illustrates that local agents are tasked with exploring the environment and collecting data for the central server without conducting any local training. This arrangement grants the server access to all data, thereby compromising the privacy safeguards in FRL. In contrast, our work allows local agents to conduct local training and prohibits data leakage to the central server. We further discuss differences between FRL and Distributed RL in Appendix E.1.

*Poisoning in FRL* is an almost unexplored territory, except for very limited existing studies. Fan et al. (2021); Jordan et al. (2024) provide a fault-tolerance guarantee for FRL. However, their frameworks require the central server to access local environments, which is a form of privacy leakage. In contrast, our work prohibits the central server from accessing any local tasks; Anwar & Raychowdhury (2021) restricts local agents from multiple local updates in its algorithm, which is not only against the nature of RL exploration but also severely expensive in the communication costs in applications. In contrast, our poisoning method is uniquely tailored to accommodate multiple local steps, aligning with realistic settings of RL scenarios. Zhang et al. (2022); Jordan et al. (2024) present FRL methods resilient against Byzantine failure. However, Byzantine failure is considered relatively naive and weak compared with real-world attacks that can be meticulously designed. In contrast, our work addresses the gap in studying strategic malicious attacks in FRL.

Finally, our work is also related to the literature of RL in strategic settings such *stochastic games*. In stochastic games, the RL agents (players) are often selfish and optimize their own objectives, and the goal is to see whether their collective behavior results in any equilibrium outcome (Ning & Xie, 2024; Altabaa et al., 2023). However, in our poisoning FRL framework, the goal is to optimize an objective function through a central

server whose computations are poisoned by malicious RL agents. Thus, although a group of distributed agents perform RL in both settings, their objectives and learning tasks are very different.

## 1.3 Overview and Contributions

We provide an overview of the remaining content in this work as follows. To address the vulnerability of FRL to adversarial attacks, we start by proposing a theoretical framework (Sections 2 and 3) that characterizes the problem of environment poisoning in FRL as an optimization problem. We assume the presence of a suspicious agent within the federated system, which can be either a malicious client or a compromised victim. This high-risk agent is trained by perturbed observations through reward manipulation. However, the attacker does not have prior knowledge of the underlying Markov Decision Process (MDP) of the environment and can only learn it through observations. As mentioned previously, this type of attack is practical and can be easily implemented in real-world scenarios, such as buying an IoT device to participate in an FRL system and providing false signals to its sensors.

To assess this risk, we design a novel poisoning mechanism (Section 4) that targets FRL with policy-based local training. Our protocol is designed to target not only general policy-based methods but also the Actor-Critic setting, wherein the attacker trains a set of both *public* and *private* critics. The private critic calculates optimized poisoned rewards, while the public critic manipulates the coordinator's model during the training process. Notably, our poisoning protocol operates without requiring knowledge of the MDP of the environment and remains consistent with the multiple local steps setting of FRL. Furthermore, we offer a theoretical guarantee for our attack mechanism (Section 5), demonstrating that our approach can lead to a substantial drop in the performance of the FRL system. We establish a theoretical result that our attack is effective even in the most challenging scenario for the attacker.

Our method is evaluated through extensive experiments on OpenGYM environments (Brockman et al., 2016), which represent standard RL tasks across various difficulty levels such as CartPole, InvertedPendulum, LunarLander, Hopper, Walker2d, and HalfCheetah. These experiments employ different FRL backbone models. The findings conclusively illustrate that, through our attack mechanism, a malicious agent can effectively poison the entire FRL system.

**Contributions.** Our findings highlight the vulnerability of the FRL framework to local poisoning attacks, emphasizing the need for awareness of system risks during training.

- **Innovative FRL Poisoning Method**. Our method addresses *the challenges of prior RL poisoning* in two key aspects: (a) *Adherence to federated privacy code.* Our method strictly avoids accessing the environment's MDP, a code often violated by previous RL poisoning. (b) *Leveraging historical federated rounds.* Our double-critic mechanism leverages historical data from prior federated rounds, overcoming the reduced poisoning effectiveness of prior RL attacks applied to federated contexts.

- **Theoretical Analysis**. We provide theoretical evidence demonstrating the impact of our method in poisoning the system. Our main theoretical findings imply that the extent of the objective's decrease scales with the square of the attack budget. Moreover, a larger learning rate and a smaller system size increase the system's vulnerability to our attack.

- **Empirical Validation**. We validate the effectiveness of our poisoning protocol through extensive experiments targeting mainstream RL algorithms like VPG and PPO. These experiments encompass OpenGYM environments with varying difficulty levels, as detailed in Section 6. This evaluation includes comparisons with various baseline models and assessments of different (targeted and non-targeted) poisoning types.

In summary, our work indicates that when FL is applied to RL training, the systematic security risk can make FRL susceptible to poisoning and threats in applications. Consequently, the trained policy of FRL may not be entirely reliable and requires a more robust and secure protocol. Our findings highlight the potential risks of FRL under adversarial attacks and inspire future research toward developing more robust algorithms.

## 2 Preliminaries and Notations

In this section, we overview some background and notations that are necessary for introducing the concept of poisoning in FRL. We consider single-task FRL, where a number of agents work together to achieve a common task. As such, all agents are trained on the same MDP. We consider the ubiquitous on-policy training setting (Singh et al., 2000).

**MDP and RL.** An MDP is a discrete stochastic control process for decision-making (Puterman, 1990) that is defined by a tuple $M = (\mathbb{S}, \mathbb{A}, r, P, \gamma)$, where $\mathbb{S}$ is a state space, $\mathbb{A}$ is an action space, $r(\cdot) : \mathbb{S} \times \mathbb{A} \to \mathbb{R}$ is a reward function, $P(\cdot) : \mathbb{S} \times \mathbb{A} \times \mathbb{S} \to [0, 1]$ is a state transition probability function, and $\gamma \in (0, 1)$ is a discount factor. Given an MDP, the goal of RL is to find an optimal policy, $\pi(\cdot) : \mathbb{S} \to \Delta_{\mathbb{A}}$, where $\Delta_{\mathbb{A}}$ is the set of all probability distributions over the action space $\mathbb{A}$, which maximizes the expected accumulated discounted reward. As is common in the literature (Agarwal et al., 2021), we often represent a policy $\pi$ by its parametrization $\boldsymbol{\theta}$ (e.g., tabular parametrization or neural network weight parametrization). During the process, at each step $t$, the decision maker begins in some state $\mathbf{s}_t$, selects an action $\mathbf{a}_t$ according to the policy $\pi(\mathbf{s}_t)$, receives a reward $r(\mathbf{s}_t, \mathbf{a}_t)$, and transitions to the next state $\mathbf{s}_{t+1}$ with probability $P(\mathbf{s}_{t+1}|\mathbf{s}_t, \mathbf{a}_t)$. This decision-making and interaction process continues until the MDP terminates.

**Federated Reinforcement Learning (FRL)**. An FRL system consists of $n$ agents and a central server. The agents perform local training for $L$ steps and then send their updated policies to the central server. The server performs aggregation to create a central policy, which is then broadcast back to all the agents. This process is repeated for $T$ rounds, and the broadcast policy is used to initialize the next round of local training. More specifically, at each round $p \leq T$, at the *start* of local step $q \leq L$, denote the policy of each agent $i$ by its policy parameter $\boldsymbol{\theta}_{(i)}^{p,q-1}$. Following this policy, the agent interacts with its environment, gathering sequences of states, actions, and rewards: $\mathbf{O}_{(i)}^{p,q} = (\mathbf{S}_{(i)}^{p,q}, \mathbf{A}_{(i)}^{p,q}, \mathbf{R}_{(i)}^{p,q}) = ((\mathbf{s}_1, \mathbf{s}_2, \ldots), (\mathbf{a}_1, \mathbf{a}_2, \ldots), (\mathbf{r}_1, \mathbf{r}_2, \ldots))$. Based on $\mathbf{O}_{(i)}^{p,q}$, the local policy is typically updated using $\boldsymbol{\theta}_{(i)}^{p,q} = \arg\max_{\boldsymbol{\theta}} J(\boldsymbol{\theta}, \boldsymbol{\theta}_{(i)}^{p,q-1}, \mathbf{O}_{(i)}^{p,q})$, where $J$ is some objective function. After completing $L$ steps of local training, all agents update their local policies to $\{\boldsymbol{\theta}_{(i)}^{p,L}\}_{i \in [n]}$, which are then submitted to the server.[1] Subsequently, the server forms a new global policy using $\boldsymbol{\theta}_{(0)}^p := \mathcal{A}^{agg}(\{\boldsymbol{\theta}_{(i)}^{p,L}\}_{i=1}^n)$, where $\mathcal{A}^{agg}$ is an aggregation algorithm. The server broadcasts the updated global policy $\boldsymbol{\theta}_{(0)}^p$ to all agents, after which each agent $i$ updates its local policy as $\boldsymbol{\theta}_{(i)}^{p+1,0} = \boldsymbol{\theta}_{(0)}^p$, and the system proceeds to the next round $p + 1$. This process repeats until the maximum federated round $T$.

## 3 Problem Formulation

We propose a theoretical framework to conceptualize the challenge of reward poisoning attacks to FRL as a sequential optimization problem as Problem (P). Problem (P) is defined in terms of the notations and concepts introduced in Section 2, such as the local policies, global policy, and aggregation algorithms. The remainder of this section is organized as follows. In Section 3.1, we elaborate on the rationale for implementing local reward poisoning. In Sections 3.2 and 3.3, we provide a detailed explanation of the objective function, feasible domain, and constraints in Problem (P). Finally, in Section 3.4, we present the knowledge possessed by each involved party, ensuring alignment with established security protocols in FL.

### 3.1 Local Reward Poisoning

We consider a threat model targeting FRL systems through reward poisoning. This model assumes the presence of an untrustworthy participant in the system, categorized as either an internal malicious agent (typical in FL poisoning (Tolpegin et al., 2020)) or a victim contaminated by external attackers (a common scenario in RL poisoning (Zhang et al., 2020)). We consider the rewards of the suspicious agents get poisoned during local training, following established practices in RL poisoning (Huang & Zhu, 2019; Zhang et al., 2020; Rakhsha et al., 2021; Banihashem et al., 2022). For clarity, we designate the compromised agent as agent $n$. At each round $p$ and step $q$, the attacker may manipulate the reward sequence of $\mathbf{R}_{(n)}^{p,q}$, transforming it

---

[1]We distinguish the parameters related to the server by index 0.

into $\widehat{\mathbf{R}}_{(n)}^{p,q}$. Throughout this paper, we will use the notation $\widehat{\cdot}$ to indicate variables that are poisoned by the attacker.

## 3.2 Objective Function and Feasible Domain

**Objective Function**. In the optimization Problem (P), the objective $\mathcal{L}_A$ represents the loss of the attacker, which can characterize both untargeted and targeted poisoning settings. In the case of untargeted poisoning, $\mathcal{L}_A$ is the benefit of the server, typically represented by the long-term cumulated reward. In the case of targeted poisoning, $\mathcal{L}_A$ is a policy matrix distance, measuring the difference between the learned policy $\boldsymbol{\theta}_{(0)}^T$ and some targeted policy $\boldsymbol{\theta}^\dagger$. This captures the attacker's goal to manipulate the global model to align with a specifically targeted policy.

**Feasible Domain**. The objective $\mathcal{L}_A$ is minimized over manipulated rewards $\widehat{\mathbf{R}}$, which is subject to the constraint that $\widehat{\mathbf{R}}$ should closely align with the ground-truth rewards $\mathbf{R}$ (Eq. 5). Although $\mathbf{R}$ may initially seem to behave as a random variable, it is essential to note that in Problem P, we characterize the process such that once $\mathbf{R}_n^{p,q}$ is obtained and observed, it is treated as a deterministic variable. Subsequently, the attacker refers to this observed deterministic variable $\mathbf{R}_n^{p,q}$ to optimize its manipulated rewards $\widehat{\mathbf{R}}_n^{p,q}$.

$$\arg \min_{\widehat{\mathbf{R}}} \mathcal{L}_A \left( \boldsymbol{\theta}_{(0)}^T \Big| \{ \widehat{\mathbf{R}}_{(n)}^{p,q} \}_{1 \le p \le T}^{1 \le q \le L} \right) \tag{P}$$

$$\text{s.t. } \boldsymbol{\theta}_{(i)}^{p,0} = \boldsymbol{\theta}_{(0)}^{p-1}, \ \forall i \le n, \tag{1}$$

$$\boldsymbol{\theta}_{(i)}^{p,q} = \arg \max_{\boldsymbol{\theta}} J(\boldsymbol{\theta}, \boldsymbol{\theta}_{(i)}^{p,q-1}, \mathbf{O}_{(i)}^{p,q-1}), \ \forall i < n, \tag{2}$$

$$\boldsymbol{\theta}_{(n)}^{p,q} = \arg \max_{\boldsymbol{\theta}} J(\boldsymbol{\theta}, \boldsymbol{\theta}_{(n)}^{p,q-1}, \widehat{\mathbf{O}}_{(n)}^{p,q-1}), \tag{3}$$

$$\boldsymbol{\theta}_{(0)}^p = \mathcal{A}^{agg}\big( \{ \boldsymbol{\theta}_{(i)}^{p,L} \}_{i \in [n]} \big), \tag{4}$$

$$D\big( \widehat{\mathbf{R}}_{(n)}^{p,q}, \mathbf{R}_{(n)}^{p,q} \big) \le \epsilon. \tag{5}$$

## 3.3 Constraints

The constraints in optimization (P) characterize the process of poisoning in FRL, including *local initialization* (Eq. 1), *local train* (Eq. 2, 3), *attack with limited budget* (Eq. 5), and *global aggregation* (Eq. 4). A concise summary is provided in Table 1, and further details are explained below.

**Local train.** Constraints 2 and 3 outline the local update of each agent at local step $q$ in round $p$. In Constraint 2, each clean agent uses its current policy, denoted by $\boldsymbol{\theta}_{(i)}^{p,q-1}$, to roll out an observation sequence $\mathbf{O}_{(i)}^{p,q-1}$. Then, based on this observation, the agent updates its policy from $\boldsymbol{\theta}_{(i)}^{p,q-1}$ to $\boldsymbol{\theta}_{(i)}^{p,q}$ by maximizing an objective function $J(\cdot)$, defined by the agent's specific RL algorithm. Constraint 3 characterizes the update for the suspicious agent's update, akin to 2, with the distinction that the local update for a suspicious agent is based on the manipulated rewards $\widehat{\mathbf{R}}$.

**Attack budget**. Constraint 5 captures the budget constraint for the attacker, where $D(\cdot, \cdot)$ represents the distance between the perturbed (poisoned) observations and the clean observations, which is restricted by a cost budget $\epsilon$. The consideration of the attack budget is crucial. On the one hand, a limited budget constitutes a standard assumption in RL reward poisoning Huang & Zhu (2019); Rakhsha et al. (2021). On the other hand, our framework strives to encompass scenarios of both internal malicious clients and victim clients susceptible to external poisoning – From a practical standpoint, an internal malicious agent may seek to avoid detection, necessitating caution in its manipulation, while an external attacker is likely to be concerned about the costs associated with their attacks. Therefore, the inclusion of an attack budget is imperative for both scenarios.

**Global aggregation and local initialization.** Constraint 4 models the aggregation step of the central server by an aggregation algorithm $\mathcal{A}^{agg}$. This algorithm processes the models sent from agents, denoted by

Table 1: Constraints of Problem P.

| Party | Constraints |
|---|---|
| Agent $(i)$, $i < n$ (Clean agents) | .Eq. 1: Local initialization
Eq. 2: Local training |
| Agent $(n)$ (suspicious agents) | Eq. 1: Local initialization
Eq. 3: Local training
Eq. 5: Attack Budget |
| Coordinator | Eq. 4: aggregation |

$\{\boldsymbol{\theta}_{(i)}^{p-1,L}\}$, as inputs and produces the new global model $\boldsymbol{\theta}_{(0)}^{p}$ as an output. Constraint 1 stipulates that at the start of local training, all agents initialize their individual policies as the global model.

### 3.4  Information Structure

We present the knowledge of each party in the FRL system to guarantee strict adherence to privacy protection in FL. Table 2 shows the knowledge of the three parties involved in FL, namely, the coordinator, the attacker, and the agents, and clarifies the knowledge of each party to guarantee the data privacy expected from FL.

The coordinator only has knowledge of the submitted models and the global policy, while the agents only have knowledge of their local data, policy, and the broadcast global model. The attacker only has knowledge of its own observations, manipulations, model, and the broadcast global policy. This ensures that the agents' private data is kept confidential.

Table 2: Knowledge of the parties in a poisoned FRL.

| | Coordinator | Agent $(i), i < n$ | Agent $(n)$ |
|---|---|---|---|
| $\mathcal{L}_{\mathcal{A}}(\cdot)$ | | | $\checkmark$ |
| $J(\cdot)$ | | $\checkmark$ | $\checkmark$ |
| $\mathcal{A}^{agg}(\cdot)$ | $\checkmark$ | | |
| $\boldsymbol{\theta}_{(0)}^{p}$ | $\checkmark$ | $\checkmark$ | $\checkmark$ |
| $\boldsymbol{\theta}_{(i)}^{p,q}, i \neq n$ | | $\checkmark$ | |
| $\boldsymbol{\theta}_{(n)}^{p,q}$ | | | $\checkmark$ |
| $\epsilon$ | | | $\checkmark$ |
| $\mathbf{O}_{(i)}^{p,q}, i \neq n$ | | $\checkmark$ | |
| $\mathbf{O}_{(n)}^{p,q}$ | | | $\checkmark$ |
| $\widehat{\mathbf{O}}_{(n)}^{p,q}$ | | | $\checkmark$ |
| $\mathbf{R}_{(i)}^{p,q}, i \neq n$ | | $\checkmark$ | |
| $\mathbf{R}_{(n)}^{p,q}$ | | | $\checkmark$ |
| $\widehat{\mathbf{R}}_{(n)}^{p,q}$ | | | $\checkmark$ |

We refer to Appendix A for a specification of this framework to the case of Proximal Policy Optimization (Schulman et al., 2017), which provides an illustrative example, demonstrating how our framework characterizes poisoning in FRL for local actor-critic algorithms.

## 4  Method

In this section, we propose practical local reward poisoning methods for FRL. We consider two scenarios when the local RL training in the FRL system uses actor-critic methods (Algorithm 2) and policy gradient methods (Algorithm 3). We note that our reward poisoning methods can be employed for both targeted and untargeted reward poisoning, as detailed in Section 3.2, by employing the appropriate corresponding objective. As before, we use $\widehat{\cdot}$ to denote the suspicious agent's parameters communicated with the server.

### 4.1 Reward Poisoning for Actor-Critic-based FRL

**Actor-Critic mechanism.** In actor-critic algorithms (Peters & Schaal, 2008), we have a policy parameterized by $\boldsymbol{\theta}$ alongside a corresponding value function $V_{\boldsymbol{\theta}}(s)$, determined by policy $\boldsymbol{\theta}$ and state $s$. Besides the policy model, each agent incorporates a critic model $\phi_{\boldsymbol{\omega}}(\cdot)$ parameterized by $\boldsymbol{\omega}$, providing an estimation of the policy-related value function $V_{\boldsymbol{\theta}}(\mathbf{s})$, denoted by $\overline{V}(s) = \phi_{\boldsymbol{\omega}}(s)$. To simplify notations, by some abuse of notation, we refer to the critic model $\phi_{\boldsymbol{\omega}}$ by its parametrization $\boldsymbol{\omega}$ in subsequent analysis. The estimated value function $\overline{V}(s)$ generated by the critic $\boldsymbol{\omega}$ further gives an estimation of the $Q$-function as $\overline{Q}(\mathbf{s}, \mathbf{a}) = r(\mathbf{s}, \mathbf{a}) + \gamma \cdot \overline{V}(\mathbf{s}')$, where $r(\mathbf{s}, \mathbf{a})$ is the observed reward, and $\mathbf{s}'$ is the next observed state. The critic model updates itself by minimizing the temporal-difference error between its model estimation and ground-truth observation (Tesauro et al., 1995). The policy parameter $\boldsymbol{\theta}$ is updated by $\boldsymbol{\theta}^{t+1} = \arg\max_{\boldsymbol{\theta}} J(\boldsymbol{\theta}, \boldsymbol{\theta}^t, \mathbf{O}^t)$, where $J(\cdot)$ is some objective function specified by the actor-critic algorithm, $t$ is the exploration step, and $\mathbf{O}^t$ is the observed trajectory. In the following, we describe the poisoning mechanism in detail and summarize it in Algorithm 2, which uses Algorithm 1 as a subroutine.

---

**Algorithm 1** Poisoned Local Train for Actor-Critic-based FRL

---

1: **Input**: current actor $\widehat{\boldsymbol{\theta}}_{(n)}^{p,q-1}$, current private critic $\boldsymbol{\omega}_{(n)}^{p,q-1}$, current public critic $\widehat{\boldsymbol{\omega}}_{(n)}^{p,q-1}$.
2: **Output**: updated actor $\widehat{\boldsymbol{\theta}}_{(n)}^{p,q}$, updated private critic $\boldsymbol{\omega}_{(n)}^{p,q}$, updated public critic $\widehat{\boldsymbol{\omega}}_{(n)}^{p,q}$.
3: Agent $n$ uses current actor $\widehat{\boldsymbol{\theta}}_{(n)}^{p,q-1}$ to obtain ground-truth observation $\mathbf{O}_{(n)}^{p,q}$,
4: computes true objective $J_{(n)}^{p,q}$ using ground-truth observation $\mathbf{O}_{(n)}^{p,q}$ and private critic $\boldsymbol{\omega}_{(n)}^{p,q-1}$,
5: poisons rewards as $\widehat{\mathbf{R}}_{(n)}^{p,q}$ using true objective $J_{(n)}^{p,q}$ and budget $\epsilon$ by Eq. 6,
6: obtains poisoned objective $\widehat{J}_{(n)}^{p,q}$ with poisoned observation $\widehat{\mathbf{O}}_{(n)}^{p,q}$ and public critic $\widehat{\boldsymbol{\omega}}_{(n)}^{p,q-1}$,
7: updates poisoned actor to $\widehat{\boldsymbol{\theta}}_{(n)}^{p,q}$ with poisoned objective $\widehat{J}_{(n)}^{p,q}$,
8: updates public critic to $\widehat{\boldsymbol{\omega}}_{(n)}^{p,q}$ with poisoned observation $\widehat{\mathbf{O}}_{(n)}^{p,q}$,
9: updates private critic to $\boldsymbol{\omega}_{(n)}^{p,q}$ with true observation $\mathbf{O}_{(n)}^{p,q}$.

---

**Public and private critics.** The term "public critic" implies that the agent uses this critic for communication with the server (Line 23), while "private critic" signifies that the agent retains this critic solely for local computation purposes (Line 4). To manipulate the global model by training (Line 8) and submitting (Line 23) a *poisoned public critic*, the attacker also undergoes training for an *unpoisoned private critic* (Line 9). This approach allows the attacker to possess knowledge of a true estimation of the value function (Line 4) to make optimal poisoning decisions (Line 5).

We denote the pair of public and private critics respectively as $\widehat{\boldsymbol{\omega}}_{(n)}^{p,q}$ and $\boldsymbol{\omega}_{(n)}^{p,q}$. The private critic $\boldsymbol{\omega}_{(n)}^{p,q}$ is trained using ground-truth rewards $\mathbf{R}_{(n)}^{p,q}$ (Line 9), and then the attacker harnesses the private critic $\boldsymbol{\omega}_{(n)}^{p,q}$ to obtain poisoned rewards $\widehat{\mathbf{R}}_{(n)}^{p,q}$ (Line 5). $\widehat{\mathbf{R}}_{(n)}^{p,q}$ is then utilized to train the public critic $\widehat{\boldsymbol{\omega}}_{(n)}^{p,q}$ (Line 8). Due to the different goals and training methods of the pair of critics, their initialization also differs. At the beginning of a new round of local training, the agent initializes its public critic with the broadcast global model (i.e., $\widehat{\boldsymbol{\omega}}_{(n)}^{p,0} = \boldsymbol{\omega}_{(0)}^{p-1}$ in Line 10), while inherits its private critic from the end of last round of training (i.e., $\boldsymbol{\omega}_{(n)}^{p,0} = \boldsymbol{\omega}_{(n)}^{p-1,L}$ in Line 10).

**Reward poisoning and actor update.** The attacker aims to minimize the objective function $J$ of the policy model (the actor) by poisoning its local rewards. When deciding how to poison the rewards, the attacker should use the unpoisoned objective $J$ given by the ground-truth rewards $\mathbf{R}$ and private critic $\boldsymbol{\omega}$ (Line 4) to obtain a true estimation and make a right poisoning decision (Line 5). Concretely, the attacker minimizes the original objective $J$ with respect to $\mathbf{R}$: for each round $p \leq T$ and local step $q \leq L$, the corrupted agent $n$ interacts with the environment and obtains the ground-truth observation $\mathbf{O}_{(n)}^{p,q}$. The attacker computes the unpoisoned objective $J_{(n)}^{p,q}$ using true rewards $\mathbf{R}_{(n)}^{p,q}$ and private critic $\boldsymbol{\omega}_{(n)}^{p,q}$ (Line 4). Then, the attacker

---

**Algorithm 2** Reward Poisoning for Actor-Critic-based FRL

---

1: **Input**: max federated rounds $T$, max local episodes $L$, number of agents $n$, poisoning budget $\epsilon$, aggregation algorithm $\mathcal{A}^{agg}$, actor-critic objective function $J$.
2: **Output**: server's actor model $\boldsymbol{\theta}_{(0)}^T$ and critic model $\boldsymbol{\omega}_{(0)}^T$.
3: Initialize the server's actor model $\boldsymbol{\theta}_{(0)}^0$ and critic model $\boldsymbol{\omega}_{(0)}^0$.
4: **for** $p = 1$ **to** $T$ **do**
5:     **for** $i = 1$ **to** $n$ **do**
6:         **if** $i \neq n$ **then**
7:             Agent $i$ initializes local actor and critic: $\boldsymbol{\theta}_{(i)}^{p,0} \leftarrow \boldsymbol{\theta}_{(0)}^{p-1}$, $\boldsymbol{\omega}_{(i)}^{p,0} \leftarrow \boldsymbol{\omega}_{(0)}^{p-1}$.
8:         **else**
9:             Agent $n$ initializes local actor $\widehat{\boldsymbol{\theta}}_{(n)}^{p,0} \leftarrow \boldsymbol{\theta}_{(0)}^{p-1}$ ,
10:             initializes local private critic as $\boldsymbol{\omega}_{(n)}^{p,0} \leftarrow \boldsymbol{\omega}_{(n)}^{p-1,L}$ and local public critic as $\widehat{\boldsymbol{\omega}}_{(n)}^{p,0} \leftarrow \boldsymbol{\omega}_{(0)}^{p-1}$.
11:         **end if**
12:         **for** $q = 1$ **to** $L$ **do**
13:             **if** $i \neq n$ **then**
14:                 Agent $i$ uses local actor $\boldsymbol{\theta}_{(i)}^{p,q-1}$ to obtain observation $\mathbf{O}_{(i)}^{p,q}$,
15:                 computes objective $J_{(i)}^{p,q}$ with observation $\mathbf{O}_{(i)}^{p,q}$ and local critic $\boldsymbol{\omega}_{(i)}^{p,q-1}$,
16:                 updates local actor to $\boldsymbol{\theta}_{(i)}^{p,q}$ with $J_{(i)}^{p,q}$, and updates local critic to $\boldsymbol{\omega}_{(i)}^{p,q}$ with $\mathbf{O}_{(i)}^{p,q}$.
17:             **else**
18:                 Agent $n$ runs Algorithm 1 (Poisoned Local Train for Actor-Critic-based FRL).
19:             **end if**
20:         **end for**
21:     **end for**
22:     Central server aggregates global actor $\boldsymbol{\theta}_{(0)}^p = \mathcal{A}^{agg}\big(\boldsymbol{\theta}_{(1)}^{p,L}, ..., \boldsymbol{\theta}_{(n-1)}^{p,L}, \widehat{\boldsymbol{\theta}}_{(n)}^{p,L}\big)$,
23:     Central server aggregates global critic $\boldsymbol{\omega}_{(0)}^p = \mathcal{A}^{agg}\big(\boldsymbol{\omega}_{(1)}^{p,L}, ..., \boldsymbol{\omega}_{(n-1)}^{p,L}, \widehat{\boldsymbol{\omega}}_{(n)}^{p,L}\big)$.
24: **end for**

---

poisons the reward according to

$$\widehat{\mathbf{R}}_{(n)}^{p,q} \leftarrow \mathbf{R}_{(n)}^{p,q} - \epsilon \cdot \frac{\nabla_{\mathbf{R}} J_{(n)}^{p,q}}{\|\nabla_{\mathbf{R}} J_{(n)}^{p,q}\|}, \tag{6}$$

which guarantees that the manipulation power is within the attack budget $\epsilon$. After the attacker has generated poisoned rewards $\widehat{\mathbf{R}}_{(n)}^{p,q}$ (Line 5), the actor is updated according to the poisoned objective $\widehat{J}_{(n)}^{p,q}$ (Line 7).

### 4.2 Reward Poisoning for Policy-Gradient-based FRL

In Policy Gradient (PG) algorithms (Silver et al., 2014), agents do not require a critic model. Therefore, we have adapted the poisoning method described in Section 4.1 that uses Actor-Critic for agents' local RL algorithm. The overall procedure is outlined in Algorithm 3. The attacker's goal is still to minimize the objective function $J$ of the policy model. Since there is no critic, the agent directly calculates $J$ from the observed $\mathbf{O}$ (Lines 14 and 15), and uses this information to decide how to poison the rewards to $\widehat{\mathbf{R}}$ by Eq. 6 (Line 16). The policy is then updated (Line 18) with the poisoned objective $\widehat{J}$ calculated by the poisoned rewards (Line 17).

## 5 A Theoretical Performance Guarantee

In this section, we prove that our poisoning method can strictly decrease the global objective under certain assumptions. We begin by considering the following assumptions, which is standard for theoretical analysis in the existing FRL literature (Jordan et al., 2024; Zhang et al., 2022; Fan et al., 2021).

---

**Algorithm 3** Reward Poisoning for Policy Gradient-based FRL
___
1: **Input**: max federated rounds $T$, max local episodes $L$, number of agents $n$, poisoning budget $\epsilon$, aggregation algorithm $\mathcal{A}^{agg}$, policy gradient objective function $J$.
2: **Output**: server's policy $\boldsymbol{\theta}_{(0)}^{T}$.
3: Initialize the server's policy $\boldsymbol{\theta}_{(0)}^{0}$.
4: **for** $p = 1$ **to** $T$ **do**
5:     **for** $i = 1$ **to** $n$ **do**
6:         **for** $q = 1$ **to** $L$ **do**
7:             **if** $i \neq n$ **then**
8:                 Initialize local policy $\boldsymbol{\theta}_{(i)}^{p,0} \leftarrow \boldsymbol{\theta}_{(0)}^{p-1}$
9:                 Interact with environment and obtain $\mathbf{O}_{(i)}^{p,q}$
10:                Compute $J_{(i)}^{p,q}$ with $\mathbf{O}_{(i)}^{p,q}$
11:                Update $\boldsymbol{\theta}_{(i)}^{p,q}$ with $J_{(i)}^{p,q}$
12:             **else**
13:                 Initialize local policy $\boldsymbol{\theta}_{(n)}^{p,0} \leftarrow \boldsymbol{\theta}_{(0)}^{p-1}$
14:                 Interact with environment and obtain clean observation $\mathbf{O}_{(n)}^{p,q}$
15:                 Compute clean objective $J_{(n)}^{p,q}$ with the clean observation $\mathbf{O}_{(n)}^{p,q}$
16:                 Poison reward as $\widehat{\mathbf{R}}_{(n)}^{p,q}$ using the clean objective $J_{(n)}^{p,q}$ and the clean observation $\mathbf{O}_{(n)}^{p,q}$ by Eq. 6
17:                 Obtain poisoned objective $\widehat{J}_{(n}^{p,q}$ with poisoned observation $\widehat{\mathbf{O}}_{(n)}^{p,q}$
18:                 Update local policy $\widehat{\boldsymbol{\theta}}_{(n)}^{p,q}$ with the poisoned objective $\widehat{J}_{(n)}^{p,q}$
19:             **end if**
20:         **end for**
21:     **end for**
22:     $\boldsymbol{\theta}_{(0)}^{p} = \mathcal{A}^{agg}\big(\boldsymbol{\theta}_{(1)}^{p,L}, ..., \boldsymbol{\theta}_{(n-1)}^{p,L}, \widehat{\boldsymbol{\theta}}_{(n)}^{p,L}\big)$
23: **end for**
___

**Assumption 1** (Single-step local training). *We assume that for each round, all local agents only apply single-step local training, i.e., $L = 1$.*

**Clean local reward sequence function.** Let us denote the dimensions of $\boldsymbol{\theta}$ and $\mathbf{r}$ by $d_{\boldsymbol{\theta}}$ and $d_{\mathbf{r}}$, respectively. Given a policy $\boldsymbol{\theta}$, define $r_{(i)}(\cdot) : \mathbb{R}^{d_{\boldsymbol{\theta}}} \to \mathbb{R}^{d_{\mathbf{r}}}$ as the function of the reward sequence generated by the agent interacting with the local environment $i$ under policy $\boldsymbol{\theta}$. Under Assumption 1, denote the reward sequence observed by agent $i$ during round $p$ by $\mathbf{r}_{(i)}^{p}$. Then, we have $\mathbf{r}_{(i)}^{p} = r_{(i)}(\boldsymbol{\theta}_{(0)}^{p-1})$.

**Clean local objective function.** With slight abuse of notations, let $J(\boldsymbol{\theta}; \mathbf{r})$ denote the local RL objective function given the policy $\boldsymbol{\theta}$ and the reward sequence $\mathbf{r}$, where we have $J : \mathbb{R}^{d_{\boldsymbol{\theta}} \times d_{\mathbf{r}}} \to \mathbb{R}$. According to the mechanism of FRL, we further define the objective of local agent $i$ given an initialized policy $\boldsymbol{\theta}$ as $J_{(i)}(\boldsymbol{\theta}) := J(\boldsymbol{\theta}; r_{(i)}(\boldsymbol{\theta}))$. If we denote the objective of local agent $i$ at the *start* of round $p$ by $J_{(i)}^{p}$, we have

$$J_{(i)}^{p} = J(\boldsymbol{\theta}_{(0)}^{p-1}; r_{(i)}(\boldsymbol{\theta}_{(0)}^{p-1})) = J(\boldsymbol{\theta}_{(0)}^{p-1}; \mathbf{r}_{(i)}^{p}).$$

**Poisoned local reward sequence and local objective.** Under Assumption 1, agent $n$ is poisoned. According to our algorithms 2 and 3, we have $\hat{\mathbf{r}}_{(n)}^{p} = \mathbf{r}_{(n)}^{p} - \epsilon \cdot \vec{e}(\nabla_{\mathbf{r}_{(n)}^{p}} J_{(n)}^{p})$, where $\vec{e}(\boldsymbol{v}) := \frac{\boldsymbol{v}}{\|\boldsymbol{v}\|}$ for any vector $\boldsymbol{v}$. Therefore, the poisoned objective is given by $\hat{J}_{(n)}^{p} = J(\boldsymbol{\theta}_{(0)}^{p-1}; \hat{\mathbf{r}}_{(n)}^{p})$.

**Local policy update function.** Under Assumption 1, each agent performs only a one-step local update. Denote the update rate as $\lambda_{\boldsymbol{\theta}} \in \mathbb{R}$. Since only agent $n$ is poisoned, we have

$$\hat{\boldsymbol{\theta}}_{(n)}^{p} = \boldsymbol{\theta}_{(0)}^{p-1} + \lambda_{\boldsymbol{\theta}} \cdot \nabla_{\boldsymbol{\theta}_{(0)}^{p-1}} \hat{J}_{(n)}^{p}.$$

**Assumption 2** (Central Aggregation). *Suppose the server updates global policy by the conventional FedAVG settings, i.e., $\boldsymbol{\theta}_{(0)}^p = \frac{1}{n}\sum_{i\in[n]}\boldsymbol{\theta}_{(i)}^p$ for clean training. In particular, when agent $n$ is poisoned, we have $\hat{\boldsymbol{\theta}}_{(0)}^p = \frac{1}{n}(\sum_{i\in[n-1]}\boldsymbol{\theta}_{(i)}^p + \hat{\boldsymbol{\theta}}_{(n)}^p)$.*

In fact, the aggregation method described in Assumption 2 aligns with common practices found in established FRL literature (Xie & Song, 2023b; Zhu & Gong, 2023).

**Global objective function.** Under the conventional setting of FL, given model $\boldsymbol{\theta}$, the global objective is defined as $J_{(0)}(\boldsymbol{\theta}) := \frac{1}{n}\sum_{i=1}^{n}J_{(i)}(\boldsymbol{\theta})$ (McMahan et al., 2017). In FRL, at the *end* of round $p$, the clean federated objective is given by $J_{(0)}^p = J_{(0)}(\boldsymbol{\theta}_{(0)}^p)$. When agent $n$ is poisoned, we have $\hat{J}_{(0)}^p = J_{(0)}(\hat{\boldsymbol{\theta}}_{(0)}^p)$.

**Assumption 3** (Objective smoothness). *We assume that $J_{(0)}^p$ is differentiable with respect to $\mathbf{r}$ and $\boldsymbol{\theta}$ almost everywhere, and $J_{(0)}^p$ is $L_{\mathbf{r}}-$smooth with respect to $\mathbf{r}_{(n)}^p$.*

Assumption 3 is well-founded, aligning with prevalent local RL objectives. We provide two illustrative examples: a) Discounted Cumulative Reward (Kaelbling et al., 1996): In this case, we have $J_{(i)}^p := \boldsymbol{\gamma} \cdot \mathbf{r}_{(i)}^p$, where $\boldsymbol{\gamma} := (\gamma^0, \gamma^1, \gamma^2, ..., \gamma^{d_{\mathbf{r}}})$ is the discount vector. Here, $\gamma \in (0, 1)$ is a discount factor, and $d_{\mathbf{r}}$ represents the cardinality of $\mathbf{r}_{(i)}^p$. b) Average Reward (Kaelbling et al., 1996): In this case, the local objective is expressed as $J_{(i)}^p := (\vec{\mathbf{1}} \cdot \mathbf{r}_{(i)}^p)/d_{\mathbf{r}}$, where $d_{\mathbf{r}}$ denotes the cardinality of $\mathbf{r}_{(i)}^p$, and $\vec{\mathbf{1}} := (1, 1, ..., 1) \in \mathbb{R}_{d_{\mathbf{r}}}$ stands for an all-one vector. We note that the primary purpose of Assumption 3 is to facilitate our theoretical analysis. However, in our numerical experiments (Section 6.1), we demonstrate the effectiveness of our proposed attack even if this assumption fails to hold (e.g., when the states/actions/rewards are discrete).

**Assumption 4** (Worst-Case). *We assume the attacker is only able to attack in the latest round.*

**Remark 5.** *Assumption 4 is designed to encapsulate the most challenging scenario for the attacker, where manipulation is restricted solely to the latest round. The proven guarantee under this assumption serves as an indicator of the effectiveness of our attack method in more general and lenient scenarios, wherein the attacker has the ability to poison the system over multiple rounds. Furthermore, our experiments showcase the effectiveness of the attack when deployed over multiple rounds (refer to Section 6).*

We are now ready to state our main theoretical result, whose proof is deferred to Appendix A.

**Theorem 6.** *Let Assumptions 1, 2, 3, and 4 hold. Suppose that all agents are updated cleanly at the first $p-1$ rounds, and at round $p$, agent $n$ is poisoned. Let $\epsilon_+ := \frac{2\lambda_{\boldsymbol{\theta}}B}{nL_{\mathbf{r}}}$, where $B$ is a scalar defined as*

$$B = [\nabla_{\boldsymbol{\theta}'}J_{(0)}(\boldsymbol{\theta}')]^\top \cdot [\nabla_{\mathbf{r},\boldsymbol{\theta}}J(\boldsymbol{\theta},\mathbf{r}) \cdot \vec{\boldsymbol{e}}(\nabla_{\mathbf{r}}J(\boldsymbol{\theta},\mathbf{r}))\Big|_{\substack{\boldsymbol{\theta}=\boldsymbol{\theta}_{(0)}^{p-1}\\\boldsymbol{\theta}'=\boldsymbol{\theta}_{(0)}^{p}\\\mathbf{r}=\mathbf{r}_{(n)}^{p}}}.$$

*Then, for $B > 0$ and $\epsilon < \epsilon_+$, we have $\hat{J}_{(0)}^p \leq J_{(0)}^p - \alpha$, where $\alpha \in (0, \frac{\epsilon_+^2}{8}]$.*

**Remark 7.** *Theorem 6 asserts that the gap $\alpha$ is strictly positive, and its upper bound increases proportionally with $\epsilon_+^2$, which is the square of the upper limit of the attack budget. With a small poison budget $\epsilon$, we can guarantee that the poisoned global objective $\hat{J}_{(0)}^p$ is strictly smaller than the clean global objective $J_{(0)}^p$, and a higher attack budget $\epsilon$ can indicate a greater decrease in the global objective. Similarly, a larger local learning rate $\lambda_{\boldsymbol{\theta}}$ or fewer agents can increase $\epsilon_+$, indicating a stronger objective decrease $\alpha$. In Appendix A, we not only explore a practical scenario ensuring $B > 0$ but also highlight that a higher learning rate $\lambda_{\boldsymbol{\theta}}$ enhances the likelihood of $B > 0$.*

## 6 Numerical Experiments

In this section, we conduct a series of experiments to evaluate the effectiveness of our poisoning method on the FRL system. Our results show that the proposed poisoning method can effectively reduce the mean episode reward of the server in the FRL system. Additionally, our poisoning protocol does not require knowledge of the MDP of the environment and is consistent with the multiple local steps setting of FRL.

## 6.1 Environments

**Nature of FRL**: Following existing FRL literatures (Jordan et al., 2024; Zhang et al., 2022), we implement experiments on various *OpenAI Gym* environments (Brockman et al., 2016) with increasing complexity, including *CartPole*, *Inverted Pendulum*, *Hopper*, *Lunar Lander*, and *Half Cheetah*, all of which are standard RL task environments. The selection of these datasets reflects the inherent nature of FRL, where the focus is on solving RL tasks, and FL serves as a versatile toolbox.

It is worth noting that environments such as *Inverted Pendulum*, *Hopper*, *Half Cheetah* feature *continuous reward spaces*, aligning with our theoretical analysis (Assumption 3). Additionally, we incorporate environments such as *Cartpole* and *Lunar Landar*, providing *discrete reward spaces*, to demonstrate the applicability of our attack in diverse scenarios of reward space.

## 6.2 Backbone Model

We explain our backbone framework by describing both its local RL model and its global FL mechanism.

**Local RL models.** Our local training backbones cover both the Policy Gradient method (corresponding to Algorithm 3) and the Actor-Critic method (corresponding to Algorithm 2), maintaining consistency with our attack methods in Section 4. For the Policy Gradient backbone, we opt for the conventional Vanilla Policy Gradient (VPG) (Sutton et al., 1999). For the Actor-Critic backbone, we choose a widely applicable model, Proximal Policy Optimization (PPO) (Schulman et al., 2017). In Appendix E.2, we discuss broader RL backbones and justify that VPG and PPO maintain accessibility and practicality while delivering competitive performance.

**Federated mechanism.** We adhere to a prevalent practice observed in existing FRL literature (Xie & Song, 2023b; Zhu & Gong, 2023), where the central server aggregates the models submitted by local agents by taking the average at the end of each communication round and then broadcasts the new global model, which is used by the local agents as initialization for the next round of local training. This federated mechanism aligns with the settings in our theoretical analysis (Assumption 2).

## 6.3 Baselines

We evaluate our methods against diverse baselines, including *standard FRL*, *poisoned FRL*, and *robust FRL*.

**Standard FRL**. In this setting, agents undergo regular training without any poisoning, adhering strictly to the conventional rules of local updates and federated communications. The baseline comprises both VPG-based standard FRL (Algorithm 5) and PPO-based standard FRL (Algorithm 4). We refer to Appendix C for a detailed description of those algorithms.

**Poisoned FRL**. To the best of our knowledge, no existing FRL method designed for poisoning matches the criteria of a standard and reasonable FRL setting, encompassing multiple local steps and data security code (refer to Section 1.2). Therefore, we establish two baselines for poisoned FRL: 1) the federated version of prior RL poisoning methods designed for single-agent systems, and 2) a randomized attack. Further details are provided below:

1. **Federated version of prior RL poisoning**. As one of our baselines, we include the federated extension of VA2C-P (Sun et al., 2020), a prior RL poisoning method. This choice is due to VA2C-P's compatibility with our reward-manipulating scheme and its adherence to the knowledge limitation that the attacker does not have access to the environment's MDP. In contrast, other prior RL poisoning methods either present compatibility issues or violate this knowledge limitation, as discussed in Section 1.1 on the challenges of applying prior RL poisoning to FRL.

2. **Randomized Poisoning**. In this attack, Eq. 6 is replaced by

$$\widehat{\mathbf{R}}^{p,q}_{(j)} \leftarrow \mathbf{R}^{p,q}_{(j)} - \epsilon \cdot \mathbf{x}, \tag{7}$$

where $\mathbf{x}$ is a vector with the same cardinality as $\mathbf{R}$, denoted as $|\mathbf{x}| = |\mathbf{R}_{(j)}^{p,q}|$. Each index-wise value in $\mathbf{x}$ follows a uniform distribution, such that $\mathbf{x} = (\mathbf{x}_1, \mathbf{x}_2, ..., \mathbf{x}_{|\mathbf{R}|})$, where $\mathbf{x}_i \sim U(0,1)$ for all $i \leq |\mathbf{R}|$. Here, $U(a,b)$ represents the uniform distribution in the range from $a$ to $b$. The detailed algorithm is provided in Appendix C.2.

**Robust FRL.** To our knowledge, no existing robust FRL method aligns with the criteria of a standard and reasonable FRL setting (refer to Section 1.2). Therefore, we adopt a standard robust mechanism inspired by FL, where the aggregation is executed by re-weighting with the clients' reliability (Fu et al., 2019; Tahmasebian et al., 2022; Wan & Chen, 2021). The detailed *defense* algorithm is outlined in Algorithm 6.

## 6.4 Experimental Settings

**Evaluation matrix.** The evaluation metrics can vary based on the type of poisoning. For *untargeted poisoning*, we evaluate the performance of these methods by measuring the mean-episode reward of the central model, which is calculated based on 100 test episodes at the end of each federated round. For *targeted poisoning*, we measure the similarity between learned policy and targeted policy. Moreover, for *discrete action space*, we calculate the proportion of target actions among all actions, while for *continuous action space*, we collect 1–scaled distance. Under both measurements, a higher value indicates a closer learned policy to the target policy. By default, our assumed poisoning type is untargeted unless stated otherwise.

**Attack Budget**. We have two configurations for the attack budget: a) *Small Budget*. We set $\epsilon = 1$ to simulate a small-budget scenario where the attacker is cautious and cost-conscious. We choose $\epsilon = 1$ as it represents the maximum possible value gained by one action move in the CartPole and Inverted Pendulum environments. It is important to note that $\epsilon = 1$ is generally much smaller than the maximum possible reward per action in the other environments (i.e., 100 for *Lunar Lander*). b) *Large Budget*. For a large-budget scenario where the attacker is bold and almost indifferent to costs, we use $\epsilon = 100$. In instances without specific notation, we refer to a small-budget attack.

**Hyper-parameters**. For both VPG and PPO settings, we let the malicious agent attack the reward with a budget of $\epsilon = 1$. There are 200 total communication rounds, and all agents run 5 local steps in each communication round. The number of poisoned agents is set to one if it is not specifically mentioned. For all experiments, we average the results over 5 random seeds.[2]

## 6.5 Empirical Results

We present experimental results that demonstrate the effectiveness of our proposed poisoning method. We compare its performance with various baselines outlined in Section 6.3, including standard FRL (Section 6.5.1), poisoned FRL (Section 6.5.2), and robust FRL (Section 6.5.2). The Experiments cover diverse backbone models (Section 6.2) and environments (Section 6.1), consistently demonstrating superior performance. In addition to evaluating the effectiveness of untargeted poisoning, we further validate our method through targeted poisoning (Section 6.5.3). Furthermore, we demonstrate the adaptability of our approach in scenarios involving multiple suspicious agents (Section 6.5.3). The results affirm that our method is both practical and detrimental for real-world applications. To enhance clarity, we have smoothed the plots by preserving values from the first 5 rounds and subsequently applying a moving average with a window size of 10 rounds.

### 6.5.1 Comparison with Standard FRL

We conducted experiments to poison standard VPG-based FRL (Fig. 1) and PPO-based FRL (Fig. 2). The backbones for these experiments were constructed as outlined in Section 6.2, utilizing Algorithms 4 and 5.

**Poison standard VPG-based FRL**. We present our results in Fig. 1. We see that across several environments, a single attacker successfully poisons a system of up to 3 to 4 agents using our method. We only present the results of systems with maximum sizes that our method can poison. For instance, in the first plot of Fig. 1, the system size is four, indicating that our method can poison a VPG-based FRL with

---

[2]We refer to Appendix D.1.1 for additional settings.

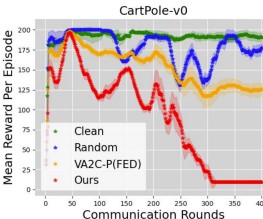 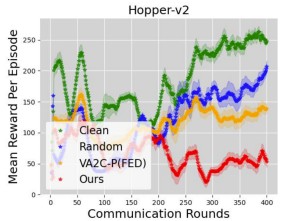 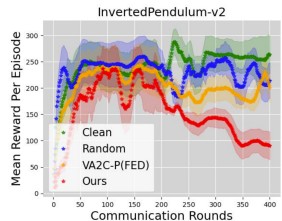 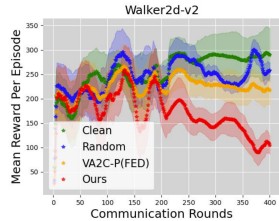

Figure 1: **Poison VPG-based FRL**. The red line, labeled *poison*, shows the performance of our adversarial attack, while the green line, labeled *clean*, represents the performance of the standard FRL system. The blue line, labeled *rand*, denotes a random poisoning baseline. The yellow line, labeled *VA2C-P(Fed)*, illustrates the federated version of VA2C-P, an existing poisoning method for single-agent RL. We specifically highlight the results for the *largest* system sizes that our method can successfully attack: 4 for *CartPole* and *Hopper*, and 3 for *Inverted Pendulum* and *Walker*.

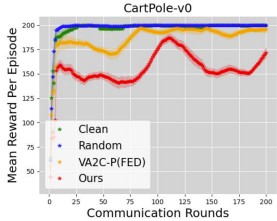 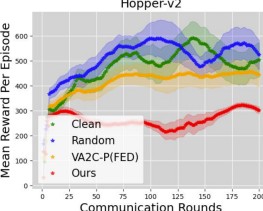 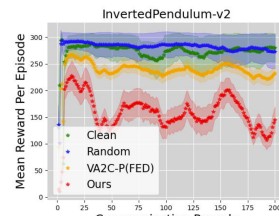 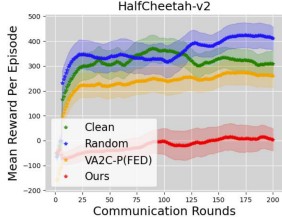

Figure 2: **Poison PPO-based FRL**. The annotation is the same as Fig. 1. We specifically showcase results for the largest system size that our method can successfully attack. The maximum system size is 3 for *Inverted Pendulum* and 4 for the others.

the number of agents equaling $1, 2, 3, 4$. However, our method would fail when the system size is 5. Across environments of varying difficulty levels, we consistently observe a significant performance gap between clean training (the green line) and training under our poisoning attack (the red line) as the number of training rounds increases. This proves the capability of our VPG attack method in poisoning federated systems.

**Poison standard PPO-based FRL.** Similar to the VPG-based FRL case, we report those systems with maximum possible size that our method can poison. Attacking a PPO-based FRL system (Fig. 2) lends itself to a much different dynamic because the separate value parameters allow all agents to learn more effectively, and thus, we expect both clean and malicious agents to be more successful in their opposite goals. We observe a consistent performance gap emerging more quickly, with fewer federated rounds, compared to the VPG-based FRL system (Fig 1). This indicates that our method is particularly effective in PPO-based FRL systems, benefiting from the double-critic protocol specifically designed for actor-critic backbones.

### 6.5.2   Comparison with Poisoned FRL and Robust FRL

**Poisoned FRL**. We conduct experiments on random attacks against standard VPG-based FRL and PPO-based FRL. We outline two poisoning baselines, a federated version of a prior RL poisoning method (VA2C-P) and a randomized attack in Section 6.3, with detailed algorithms provided in Section C.2. For VPG-based FRL, we observe that our method (Fig. 1, the red line) significantly reduces rewards of various environments compared to both the federated version of VA2C-P poisoning (Fig. 1, the yellow line) and the random attack (Fig. 1, blue line). For PPO-based FRL, both the federated version of VA2C-P (Fig. 2, the yellow line) and the random attack (Fig. 2, the blue line) perform very poorly in poisoning, as they are overwhelmed by the strength of the clean agents, leading to rewards that remain close to clean training (Fig. 2, the green line) over time, considering performance variance. In contrast, our poisoning (Fig. 2, the red line) significantly harms the system performance, outperforming the poisoning baselines.

**Robust FRL.** The selection of the robust FRL baseline is detailed in Section 6.3. Experiments are conducted following settings in Section 6.4 and D.1.1. Results are presented in Fig. 3. It is noteworthy that, although the defense mechanism can successfully mitigate the harm of our poisoning in relatively simple environments (*CartPole*, *Hopper*, *Walker2d*), for complicated environments (*Half Cheetah*), which are closer to real-world scenarios, the defense mechanism fails against our poisoning. Thus, our results consolidate that this kind of poisoning is harmful in real-world applications.

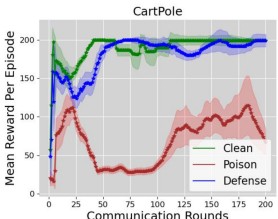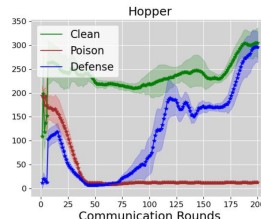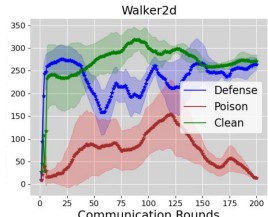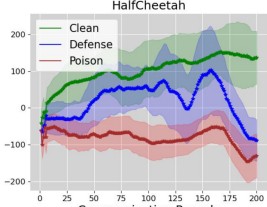

Figure 3: **Poison Robust FRL**. The red dashed line, labeled as *Poison*, depicts the performance of our adversarial attack in a standard FRL system. The green dashed line, labeled as *Clean*, represents the standard FRL system's performance. The blue dashed line, labeled as *Defense*, showcases the performance of our attack in a robust FRL system.

### 6.5.3 Additional Results

All additional results are deferred to Appendix D.2, including:

(a) **Multiple Poisoned Agents**. Fig 5 illustrates the method's applicability to scenarios with multiple poisoned agents. The attack performance remains stable, given a consistent ratio of poisoned agents to unpoisoned agents.

(b) **Targeted Poisoning**. Fig 6 demonstrates the effectiveness of our proposed method in targeted poisoning scenarios.

(c) **High-Budget Poisoning**. Fig 4 showcases that, with a sufficiently high attack budget, our proposed method empowers a single poisoned agent to impact an FRL system of considerable size (e.g., a system of 100 agents).

## 7 Broader Impact

The development of our poisoning method for FRL has significant implications for both research and practical applications. By highlighting the vulnerabilities inherent in FRL frameworks, our work raises critical security concerns in sensitive domains such as healthcare and finance, while guiding the design of robust policies to mitigate these risks. Additionally, our research paves the way for further exploration of adversarial poisoning and defensive strategies in FRL, emphasizing the need for resilient algorithms capable of withstanding malicious manipulation during the training process.

Moreover, our findings initiate important ethical discussions about the responsible use of federated learning technologies. As these systems become more integrated into critical reinforcement learning applications, it is essential to establish guidelines that ensure their deployment is both secure and beneficial. By highlighting potential poisoning risks, we encourage the adoption of security measures that not only protect data privacy but also enhance public trust in FRL systems.

## 8 Conclusions

In this work, we have proposed a novel method for poisoning FRL under both general policy-based and actor-critic algorithms, which can provably decrease the system's global objective. Our method is evaluated

through extensive experiments on various OpenGYM environments using popular RL models, and the results demonstrate that our method is effective in poisoning FRL systems. Our work highlights the potential risks of FRL and inspires future research to design more robust algorithms to protect FRL against poisoning attacks.

## Acknowledgments

This work is supported by the Air Force Office of Scientific Research under award number FA9550-23-1-0107 and the NSF CAREER Award under Grant No. EPCN-1944403.

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

# A  Proof of Theorem 6

**Theorem 8** (Theorem 6-restated). *Let Assumptions 1, 2, and 3 hold. Suppose that all agents are updated cleanly at the first $p-1$ rounds, and at round $p$, agent $n$ is poisoned. Let $\epsilon_+ := \frac{2\lambda_{\boldsymbol{\theta}} B}{n L_{\mathbf{r}}}$, where $B$ is defined as*

$$
B = [\nabla_{\boldsymbol{\theta}'} J_{(0)}(\boldsymbol{\theta}')]^\top \cdot [\nabla_{\mathbf{r},\boldsymbol{\theta}} J(\boldsymbol{\theta}, \mathbf{r}) \cdot \vec{e}(\nabla_r J(\boldsymbol{\theta}, \mathbf{r}))] \Big|_{\substack{\boldsymbol{\theta}=\boldsymbol{\theta}_{(0)}^{p-1} \\ \boldsymbol{\theta}'=\boldsymbol{\theta}_{(0)}^{p} \\ \mathbf{r}=\mathbf{r}_{(n)}^{p}}}.
$$

*Then, for $B > 0$ and $\epsilon < \epsilon_+$, we have $\widehat{J}_{(0)}^p < J_{(0)}^p - \alpha,$, where $\alpha \in [0, \frac{\epsilon_+^2}{8}]$.*

*Proof.* Since $J_{(0)}^p$ is $L_{\mathbf{r}}$-smooth with respect to $\mathbf{r}_{(n)}^p$, we have

$$
\begin{aligned}
\widehat{J}_{(0)}^p \leq \hat{J}_{(0)}^p &+ [\nabla_{\mathbf{r}_{(n)}^p} J_{(0)}^p]^\top \cdot [\widehat{\mathbf{r}}_{(n)}^p - \mathbf{r}_{(n)}^p] \\
&+ \frac{L_{\mathbf{r}}}{2} \|\widehat{\mathbf{r}}_{(n)}^p - \mathbf{r}_{(n)}^p\|^2.
\end{aligned} \tag{8}
$$

Moreover, we can write

$$
\begin{aligned}
[\nabla_{\mathbf{r}_{(n)}^p} J_{(0)}^p]^\top &= [\nabla_{\mathbf{r}_{(n)}^p} \frac{1}{n} \sum_{i \in [n]} J_{(i)}(\boldsymbol{\theta}_{(0)}^p)]^\top \\
&= [\nabla_{\boldsymbol{\theta}_{(0)}^p} \frac{1}{n} \sum_{i \in [n]} J_{(i)}(\boldsymbol{\theta}_{(0)}^p)]^\top \cdot [\nabla_{\mathbf{r}_{(n)}^p} \boldsymbol{\theta}_{(0)}^p],
\end{aligned} \tag{9}
$$

and

$$
\begin{aligned}
\nabla_{\mathbf{r}_{(n)}^p} \boldsymbol{\theta}_{(0)}^p &= \nabla_{\mathbf{r}_{(n)}^p} \frac{1}{n} \sum_{i \in [n]} \boldsymbol{\theta}_{(i)}^p \\
&= \frac{1}{n} \nabla_{\mathbf{r}_{(n)}^p} \boldsymbol{\theta}_{(n)}^p \\
&= \frac{1}{n} \nabla_{\mathbf{r}_{(n)}^p} [\boldsymbol{\theta}_{(0)}^{p-1} + \lambda_{\boldsymbol{\theta}} \nabla_{\boldsymbol{\theta}_{(0)}^{p-1}} J(\boldsymbol{\theta}_{(0)}^{p-1}, \mathbf{r}_{(n)}^p)] \\
&= \frac{\lambda_{\boldsymbol{\theta}}}{n} \nabla_{\mathbf{r}_{(n)}^p} \nabla_{\boldsymbol{\theta}_{(0)}^{p-1}} J(\boldsymbol{\theta}_{(0)}^{p-1}, \mathbf{r}_{(n)}^p).
\end{aligned} \tag{10}
$$

Substituting Eq. 10 into Eq. 9, we obtain

$$[\nabla_{\mathbf{r}_{(n)}^p} J_{(0)}^p]^\top = \frac{\lambda_{\boldsymbol{\theta}}}{n^2} \sum_{i \in [n]} [\nabla_{\boldsymbol{\theta}_{(0)}^p} J_{(i)}(\boldsymbol{\theta}_{(0)}^p)]^\top \cdot [\nabla_{\mathbf{r}_{(n)}^p} \nabla_{\boldsymbol{\theta}_{(0)}^{p-1}} J(\boldsymbol{\theta}_{(0)}^{p-1}, \mathbf{r}_{(n)}^p)]. \tag{11}$$

On the other hand, we have

$$\hat{\mathbf{r}}_{(n)}^p = \mathbf{r}_{(n)}^p - \epsilon \cdot \vec{\boldsymbol{e}}(\nabla_{\mathbf{r}_{(n)}^p} J_{(n)}^p). \tag{12}$$

If we combine Eq. 12 with Eq. 11, we get

$$[\nabla_{\mathbf{r}_{(n)}^p} J_{(0)}^p]^\top \cdot [\hat{\mathbf{r}}_{(n)}^p - \mathbf{r}_n^p] = -\frac{\epsilon \cdot \lambda_{\boldsymbol{\theta}}}{n^2} \sum_{i \in [n]} [\nabla_{\boldsymbol{\theta}_{(0)}^p} J_{(i)}(\boldsymbol{\theta}_{(0)}^p)]^\top \cdot [\nabla_{\mathbf{r}_{(n)}^p} \nabla_{\boldsymbol{\theta}_{(0)}^{p-1}} J(\boldsymbol{\theta}_{(0)}^{p-1}, \mathbf{r}_{(n)}^p)] \cdot \vec{\boldsymbol{e}}(\nabla_{\mathbf{r}_{(n)}^p} J_{(n)}^p). \tag{13}$$

Substituting Eq. 13 into Eq. 8 and using the fact that $\frac{L_{\mathbf{r}}}{2} \|\hat{\mathbf{r}}_{(n)}^p - \mathbf{r}_{(n)}^p\|^2 = \frac{\epsilon^2 \cdot L_{\mathbf{r}}}{2}$, we get

$$\widehat{J}_{(0)}^p \leq \hat{J}_{(0)}^p - \frac{\lambda_{\boldsymbol{\theta}} \cdot B}{n} \epsilon + \frac{L_{\mathbf{r}}}{2} \epsilon^2. \tag{14}$$

As the right-hand side of Eq. 14 is a quadratic function with respect to $\epsilon$, we get that when $B > 0$ and $0 < \epsilon < \frac{2\lambda_{\boldsymbol{\theta}} B}{n L_{\mathbf{r}}}$, it holds that $\hat{J}_{(0)}^p < J_{(0)}^p$, indicating a strict decrease in the objective of FRL being poisoned compared with clean training. In particular, the smallest bound in Eq. 14 is achieved when $\epsilon = \frac{\lambda_{\boldsymbol{\theta}} B}{n L_{\mathbf{r}}}$, implying $\widehat{J}_{(0)}^p \leq J_{(0)}^p - \frac{\lambda_{\boldsymbol{\theta}}^2 B^2}{2 L_{\mathbf{r}}^2 n^2} = J_{(0)}^p - \frac{\epsilon_+^2}{8}$. $\qquad\square$

**A case where $B > 0$.** Suppose $J(\boldsymbol{\theta}; \mathbf{r}) = \boldsymbol{\gamma}^\top \cdot \mathbf{r}$, where $\mathbf{r}$ is the reward sequence and $\boldsymbol{\gamma}$ is the discount factor vector. This objective corresponds with the typical accumulated discounted reward setting. Suppose that $r_{(n)}(\boldsymbol{\theta})$ is a differentiable function. Recall that

$$B = [\nabla_{\boldsymbol{\theta}^*} J_{(0)}(\boldsymbol{\theta}^*)]^\top \cdot [\nabla_{\mathbf{r}, \boldsymbol{\theta}} J(\boldsymbol{\theta}, \mathbf{r})] \cdot \vec{\boldsymbol{e}}(\nabla_r J(\boldsymbol{\theta}, \mathbf{r})) \Big|_{\substack{\boldsymbol{\theta} = \boldsymbol{\theta}_{(0)}^{p-1} \\ \boldsymbol{\theta}^* = \boldsymbol{\theta}_{(0)}^p \\ \mathbf{r} = \mathbf{r}_{(n)}^p}}.$$

Define

$$B_1 := \nabla_{\boldsymbol{\theta}^*} J_{(0)}(\boldsymbol{\theta}^*) \Big|_{\boldsymbol{\theta}^* = \boldsymbol{\theta}_{(0)}^p},$$

$$B_2 := \nabla_{\mathbf{r}, \boldsymbol{\theta}} J(\boldsymbol{\theta}, \mathbf{r}) \Big|_{\substack{\boldsymbol{\theta} = \boldsymbol{\theta}_{(0)}^{p-1} \\ \mathbf{r} = \mathbf{r}_{(n)}^p}},$$

$$B_3 := \vec{\boldsymbol{e}}(\nabla_r J(\boldsymbol{\theta}, \mathbf{r})) \Big|_{\substack{\boldsymbol{\theta} = \boldsymbol{\theta}_{(0)}^{p-1} \\ \mathbf{r} = \mathbf{r}_{(n)}^p}}.$$

We have $B = B_1^\top \cdot B_2 \cdot B_3$. To intuitively understand $B$, we simplify the reward $|\mathbf{r}| = 1$, where $|\cdot|$ denotes cardinality, and correspondingly $\boldsymbol{\gamma}$ is simplified to a scalar: $\gamma = 1$. Since

$$J_{(0)}(\boldsymbol{\theta}_{(0)}^p) = \frac{1}{n} \sum_{i=1}^n J_{(i)}(\boldsymbol{\theta}_{(0)}^p) = \frac{1}{n} \sum_{i=1}^n J(\boldsymbol{\theta}_{(0)}^p; r_{(i)}(\boldsymbol{\theta}_{(0)}^p)) = \frac{1}{n} \sum_{i \in [n]} r_{(i)}(\boldsymbol{\theta}_{(0)}^p).$$

we have

$$B_1 := \nabla_{\boldsymbol{\theta}^*} J_{(0)}(\boldsymbol{\theta}^*) \Big|_{\boldsymbol{\theta}^* = \boldsymbol{\theta}_{(0)}^p} = \frac{1}{n} \cdot \Big[ \sum_{i \in [n]} r'_{(i)}(\boldsymbol{\theta}_{(0)}^p) \Big].$$

Since

$$J(\boldsymbol{\theta}_{(0)}^{p-1}, \mathbf{r}_{(n)}^p) := J(\boldsymbol{\theta}_{(0)}^{p-1}, r_{(n)}(\boldsymbol{\theta}_{(0)}^{p-1})) = r_{(n)}(\boldsymbol{\theta}_{(0)}^{p-1}) = \mathbf{r}_{(n)}^p,$$

we have

$$B_2 := \nabla_{\mathbf{r},\boldsymbol{\theta}} J(\boldsymbol{\theta},\mathbf{r})\Big|_{\mathbf{r}=\mathbf{r}^p_{(n)}}^{\boldsymbol{\theta}=\boldsymbol{\theta}^{p-1}_{(0)}} = \nabla_{\mathbf{r}^p_{(n)}} \nabla_{\boldsymbol{\theta}^{p-1}_{(0)}} r_{(n)}(\boldsymbol{\theta}^{p-1}_{(0)}) = \nabla_{\mathbf{r}^p_{(n)}} \left[ [r'_{(n)}(\boldsymbol{\theta}^{p-1}_{(0)})] \right] = r''_{(n)}(\boldsymbol{\theta}^{p-1}_{(0)}) \cdot (r'_{(n)}(\boldsymbol{\theta}^{p-1}_{(0)}))^{-1},$$

$$B_3 = 1.$$

Therefore,

$$B = \left[ \frac{1}{n} \sum_{i \in [n]} r'_{(i)}(\boldsymbol{\theta}^p_{(0)}) \right] \cdot \frac{r''_{(n)}(\boldsymbol{\theta}^{p-1}_{(0)})}{r'_{(n)}(\boldsymbol{\theta}^{p-1}_{(0)})}.$$

Suppose that agents interact with the same environment, that is $r_{(i)}(\boldsymbol{\theta}) = r(\boldsymbol{\theta}), \forall i \in [n]$. Then, if $r'''(\boldsymbol{\theta}) = 0$, we get

$$\boldsymbol{\theta}^p_{(0)} = \boldsymbol{\theta}^{p-1}_{(0)} + \lambda_{\boldsymbol{\theta}} r'(\boldsymbol{\theta}^{p-1}_{(0)}),$$

and hence,

$$
\begin{aligned}
B &= \frac{r'(\boldsymbol{\theta}^p_{(0)}) r''(\boldsymbol{\theta}^{p-1}_{(0)})}{r'(\boldsymbol{\theta}^{p-1}_{(0)})} \\
&= \frac{\left[ r'(\boldsymbol{\theta}^{p-1}_{(0)}) + r''(\boldsymbol{\theta}^{p-1}_{(0)}) \lambda_{\boldsymbol{\theta}} \cdot r'(\boldsymbol{\theta}^{p-1}_{(0)}) \right] r''(\boldsymbol{\theta}^{p-1}_{(0)})}{r'(\boldsymbol{\theta}^{p-1}_{(0)})} \\
&= \left( 1 + \lambda_{\boldsymbol{\theta}} \cdot r''(\boldsymbol{\theta}^{p-1}_{(0)}) \right) r''(\boldsymbol{\theta}^{p-1}_{(0)}),
\end{aligned}
$$

which is a quadratic function w.r.t. $r''(\boldsymbol{\theta}^{p-1}_{(0)})$. Therefore, in this case we always have $B > 0$, as long as $r''(\boldsymbol{\theta}^{p-1}_{(0)}) \in (-\infty, -\lambda_{\boldsymbol{\theta}}^{-1}) \bigcup (0, +\infty)$. In particular, a higher rate $\lambda_{\boldsymbol{\theta}}$ makes it easier to achieve a positive $B$.

## B  Proximal Policy Optimization (PPO)-Specific Framework

In this appendix, we focus on a specific local RL algorithm, Proximal Policy Optimization (PPO) (Schulman et al., 2017), for the individual agents in FRL and propose a corresponding framework for poisoning. By specifying the local RL algorithm as PPO, we are able to tailor the problem formulation in Problem P and accordingly propose a targeted solution in Section 4 to poisoning PPO-specific FRL. In Section B, we introduce PPO preliminaries to specify the general variable in Problem P. Then we discuss the PPO-specific problem formulation for poisoning FRL in Section B. This will allow us to take into account the specific characteristics of the PPO algorithm when defining the problem.

### PPO Preliminaries

PPO is a popular Actor-Critic algorithm that uses a clipped surrogate objective. For agent $i$ at federated round $p$ and local episode $q$, denote the pair of state and action at its $t$-th step of rollout as $\mathbf{O}^{p,q,t}_{(i)} = (\mathbf{s}^{p,q,t}_{(i)}, \mathbf{a}^{p,q,t}_{(i)})$ and denote the $V$-function and $Q$-function defined by Bellman Equation (Baird, 1995) as $V(\cdot)$ and $Q(\cdot, \cdot)$, respectively. Then the advantage function is $A^{p,q,t}_{(i)} := Q(\mathbf{s}^{p,q,t}_{(i)}, \mathbf{a}^{p,q,t}_{(i)}) - V(\mathbf{s}^{p,q,t}_{(i)})$.

**PPO's critic.** Let us use $\bar{\cdot}$ to denote estimation. Denote PPO's critic model as $\phi(\cdot | \boldsymbol{\omega}^{p,q}_{(i)})$, where $\boldsymbol{\omega}$ is the model's weights. As with all typical actor-critic algorithms, the critic is a Value neural network to help estimate the V-value of the actor so as to further calculate the actor's objective. In PPO, the actor's objective is a clipped advantage (Eq. 15), where the advantage is estimated by the critic and observation, which can be written in the form of $\overline{A}^{p,q,t}_{(i)} = A(\boldsymbol{\omega}^{p,q}_{(i)}, \mathbf{O}^{p,q,t}_{(i)})$. The critic model updates itself by minimizing the temporal-difference error (Tesauro et al., 1995) between the estimated and observed $V$-value. We denote the critic's objective by $\delta^{p,q}_{(i)}$.

**PPO's actor.** Denote the actor model as $\pi_{\boldsymbol{\theta}}(\cdot|\mathbf{s}, \boldsymbol{\theta})$, where $\boldsymbol{\theta}$ is the model weight and $\mathbf{s}$ is some given state. To specify the general problem P to the PPO case, the clean agent's objective $J(\cdot)$ (Eq. 2) should be the PPO surrogate objective

$$J_{(i)}^{p,q} := \mathbb{E}_t \big[ \min \big( \gamma_{(i)}^{p,q,t} \cdot \overline{A}_{(i)}^{p,q,t}, c_{(i)}^{p,q,t} \cdot \overline{A}_{(i)}^{p,q,t} \big) \big], \tag{15}$$

where $\gamma_{(i)}^{p,q,t} := \pi(\mathbf{a}_{(i)}^{p,q,t}|\mathbf{s}_{(i)}^{p,q,t}, \boldsymbol{\theta}_{(i)}^{p,q})/\pi(\mathbf{a}_{(i)}^{p,q,t}|\mathbf{s}_{(i)}^{p,q,t}, \boldsymbol{\theta}_{(i)}^{p,q-1})$, and $\overline{A}_{(i)}^{p,q,t}$ is estimated based on both PPO's critic and the observation that the actor samples. Here, $c_{(i)}^{p,q,t} := \text{clip}(\gamma_{(i)}^{p,q,t}, 1-\eta, 1+\eta)$, where $\text{clip}(\cdot)$ is a clipping function parameterized by $\eta$.

**PPO-specific FRL poisoning.**

As an actor-critic algorithm, when we fit single-agent PPO into a federated framework, we assume that besides the actor model, the critic model should also be updated from individual agents to the server, then aggregated by the server and finally broadcast to the local agents at each federated round $p$. Denote the aggregated actor as $\boldsymbol{\theta}_{(0)}^p$ and the aggregated critic as $\boldsymbol{\omega}_{(0)}^p$. To specify the server's aggregation function $\mathcal{A}^{agg}$ (Eq. 4), we take a conventional paradigm in FL:

$$\boldsymbol{\theta}_{(0)}^p = \mathcal{A}^{agg}(\boldsymbol{\theta}_{(0)}^{p-1}, \{\boldsymbol{\theta}_{(i)}^{p,L}\}_{i=1}^n) \tag{16}$$

$$:= \boldsymbol{\theta}_{(0)}^{p-1} + \frac{1}{n} \sum_{i=1}^n (\boldsymbol{\theta}_{(i)}^{p,L} - \boldsymbol{\theta}_{(i)}^{p,0}) = \frac{\sum_{i=1}^n \boldsymbol{\theta}_{(i)}^{p,L}}{n}, \tag{17}$$

where $\mathcal{A}^{agg}$ aggregates the local models by adding the averaged local model update to the server's model (Bhagoji et al., 2019; Bagdasaryan et al., 2020). By substituting $\boldsymbol{\theta}_{(i)}^{p,0} = \boldsymbol{\theta}_{(0)}^{p-1}$, $A^{agg}$ is equivalent to assigning the averaged local model as the server's model (Bhagoji et al., 2019).

We set the poison cost as $D(\mathbf{R}^{p,q}, \widehat{\mathbf{R}}^{p,q}) = \|\mathbf{R}^{p,q} - \widehat{\mathbf{R}}^{p,q}\|_2$, and thereby propose the PPO-specific Problem as:

$$\underset{\widehat{\mathbf{R}}}{\arg\min} \; \mathcal{L}_A\Big(\boldsymbol{\theta}_{(0)}^T, \boldsymbol{\omega}_{(0)}^T \Big| \{\widehat{\mathbf{R}}_{(n)}^{p,q}\}_{1 \leq p \leq T}^{1 \leq q \leq L}\Big) \tag{P-PPO}$$

$$\text{s.t. } \forall 1 \leq p \leq T, \; 1 \leq q \leq L, \tag{18}$$

$$\boldsymbol{\theta}_{(i)}^{p,0} = \boldsymbol{\theta}_{(0)}^{p-1}, \forall i \leq n, \tag{19}$$

$$\boldsymbol{\omega}_{(i)}^{p,0} = \boldsymbol{\omega}_{(0)}^{p-1}, \forall i \leq n, \tag{20}$$

$$\boldsymbol{\theta}_{(i)}^{p,q} = \underset{\boldsymbol{\theta}}{\arg\max} \; J_{(i)}^{p,q}, \forall i < n, \tag{21}$$

$$\boldsymbol{\omega}_{(i)}^{p,q} = \underset{\boldsymbol{\omega}}{\arg\min} \; \delta_{(i)}^{p,q}, \forall i < n, \tag{22}$$

$$\boldsymbol{\theta}_{(n)}^{p,q} = \underset{\boldsymbol{\theta}}{\arg\max}[\widehat{J}_{(n)}^{p,q}|\widehat{\mathbf{O}}_{(n)}^{p,q}], \tag{23}$$

$$\boldsymbol{\omega}_{(n)}^{p,q} = \underset{\boldsymbol{\omega}}{\arg\min} \big[\widehat{\delta}_{(n)}^{p,q}|\widehat{\mathbf{O}}_{(n)}^{p,q}\big], \tag{24}$$

$$\boldsymbol{\theta}_{(0)}^p = \sum_{i=1}^n \boldsymbol{\theta}_{(i)}^{p,L}/n, \tag{25}$$

$$\|\mathbf{R}^{p,q} - \widehat{\mathbf{R}}^{p,q}\|_2 \leq \epsilon. \tag{26}$$

In Problem P-PPO, $\widehat{\phantom{a}}$ denotes the poisoned variables induced by $\widehat{\mathbf{O}}^{p,q}$. The constraints interpretation is similar to that in Section 3.3, except that all the equations related to $\boldsymbol{\omega}$ characterize the initialization and local training for the critic, while those related to $\boldsymbol{\theta}$ are for the actor. The constraints are summarized in Table 3.

Table 3: Constraints of Problem P-PPO.

| Party | Constraints | Interpretation |
|---|---|---|
| Agent $i$, $i \neq n$ (clean agents) | Eq.(19) | local actor Initialization |
| | Eq.(21) | local actor train |
| | Eq.(20) | Local critic initialization |
| | Eq.(22) | local critic train |
| Agent $n$ (attacker) | Eq.(19) | local actor initialization |
| | Eq.(23) | local actor train |
| | Eq.(20) | local critic initialization |
| | Eq.(24) | local critic train |
| | Eq.(26) | Attack budget |
| Coordinator | Eq.(25) | aggregation |

# C  Baseline Algorithms

In this appendix, we provide the baseline algorithms described in Section 6.3.

## C.1  Standard FRL

Standard Actor-Critic-based FRL is given in Algorithm 4, and the standard Policy-Gradient-based FRL is given in Algorithm 5.

---

**Algorithm 4** Standard Actor-Critic-based FRL

---

1: **Input**: max federated rounds $T$, max local episodes $L$, number of agents $n$, aggregation algorithm $\mathcal{A}^{agg}$, actor-critic objective function $J$.
2: **Output**: server's actor model $\boldsymbol{\theta}_{(0)}^{T}$ and critic model $\boldsymbol{\omega}_{(0)}^{T}$.
3: Initialize the server's actor model $\boldsymbol{\theta}_{(0)}^{0}$ and critic model $\boldsymbol{\omega}_{(0)}^{0}$.
4: **for** $p = 1$ **to** $T$ **do**
5:     **for** $i = 1$ **to** $n$ **do**
6:         Initialize local actor $\boldsymbol{\theta}_{(i)}^{p,0} \leftarrow \boldsymbol{\theta}_{(0)}^{p-1}$
7:         Initialize local critic $\boldsymbol{\omega}_{(i)}^{p,0} \leftarrow \boldsymbol{\omega}_{(0)}^{p-1}$
8:         **for** $q = 1$ **to** $L$ **do**
9:             Interact with environment and obtain $\mathbf{O}_{(i)}^{p,q}$
10:             Compute $J_{(i)}^{p,q}$ with $\mathbf{O}_{(i)}^{p,q}$ and $\boldsymbol{\omega}_{(i)}^{p,q-1}$
11:             Update $\boldsymbol{\theta}_{(i)}^{p,q}$ with $J_{(i)}^{p,q}$
12:             Update $\boldsymbol{\omega}_{(i)}^{p,q}$ with $\mathbf{O}_{(i)}^{p,q}$
13:         **end for**
14:     **end for**
15:     $\boldsymbol{\theta}_{(0)}^{p} = \mathcal{A}^{agg}\big(\boldsymbol{\theta}_{(1)}^{p,L}, ..., \boldsymbol{\theta}_{(n-1)}^{p,L}, \boldsymbol{\theta}_{(n)}^{p,L}\big)$
16:     $\boldsymbol{\omega}_{(0)}^{p} = \mathcal{A}^{agg}\big(\boldsymbol{\omega}_{(1)}^{p,L}, ..., \boldsymbol{\omega}_{(n-1)}^{p,L}, \boldsymbol{\omega}_{(n)}^{p,L}\big)$
17: **end for**

**Algorithm 5** Standard Policy-Gradient-based FRL

---

1: **Input**: max federated rounds $T$, max local episodes $L$, number of agents $n$, aggregation algorithm $\mathcal{A}^{agg}$, policy gradient objective function $J$.
2: **Output**: server's policy $\boldsymbol{\theta}_{(0)}^T$.
3: Initialize the server's policy $\boldsymbol{\theta}_{(0)}^0$.
4: **for** $p = 1$ **to** $T$ **do**
5:    **for** $i = 1$ **to** $n$ **do**
6:       **for** $q = 1$ **to** $L$ **do**
7:          Initialize local policy $\boldsymbol{\theta}_{(i)}^{p,0} \leftarrow \boldsymbol{\theta}_{(0)}^{p-1}$
8:          Interact with environment and obtain $\mathbf{O}_{(i)}^{p,q}$
9:          Compute $J_{(i)}^{p,q}$ with $\mathbf{O}_{(i)}^{p,q}$
10:         Update $\boldsymbol{\theta}_{(i)}^{p,q}$ with $J_{(i)}^{p,q}$
11:       **end for**
12:    **end for**
13:    $\boldsymbol{\theta}_{(0)}^p = \mathcal{A}^{agg}\left(\boldsymbol{\theta}_{(1)}^{p,L}, ..., \boldsymbol{\theta}_{(n-1)}^{p,L}, \boldsymbol{\theta}_{(n)}^{p,L}\right)$
14: **end for**

---

## C.2   Random Poisoning

Below we give random poisoning algorithms mentioned in Baselines (Section 6.3).

**Random Poisoning for Actor-Critic-based FRL**. The algorithm is implemented almost the same as Algorithm 2, except that Line 5 should be replaced with "Poison Reward as $\widehat{\mathbf{R}}_{(n)}^{p,q}$ using true objective $J_{(n)}^{p,q}$ and budget $\epsilon$ by Eq. (7)".

**Random Poisoning for Policy-Gradient-based FRL**. The algorithm is implemented almost the same as Algorithm 3, except that Line 16 should be replaced with "Poison Reward as $\widehat{\mathbf{R}}_{(n)}^{p,q}$ using true objective $J_{(n)}^{p,q}$ and budget $\epsilon$ by Eq. (7)".

## C.3   Robust FRL

To mitigate the risk of FRL being exposed to malicious agents, below we describe a defense mechanism against FRL (Algorithm (6)) that inherits from conventional FL defense mechanisms, where the aggregation is implemented by re-weighting with the clients' reliability. To adapt this robust aggregation protocol from FL to FRL, we let the clients' local RL performance be a reflection of their reliability. The aggregation process incorporates a re-weighting mechanism based on the clients' reliability, as discussed in prior work (Fu et al., 2019; Tahmasebian et al., 2022; Wan & Chen, 2021). In adapting this robust aggregation protocol from FL to FRL, we leverage the local RL performance of clients as an indicator of their reliability. To be more precise, the re-weighting is accomplished by assigning credits to each agent's policy according to its performance. To that end, the central server runs tests on each policy it receives from the agents and records the observations denoted by $\mathbf{O}_{(i)}^{p,test}$. The server then calculates the average reward $\bar{\mathbf{r}}_i^{p,test}$ for each policy by averaging the rewards in the sequence $\{\mathbf{r}_t\}_{(i)}^{p,test}$. Finally, the server normalizes the average rewards by dividing them by the sum of all averaged rewards, resulting in a set of normalized weights $c_i^{p,q}$. These weights are used to weight the average aggregation of the policies:

$$\boldsymbol{\theta}_{(0)}^p \leftarrow \sum_{i \in \mathcal{M}} c_i^{p,q} \widehat{\boldsymbol{\theta}}_{(i)}^{p,L} + \sum_{i \notin \mathcal{M}} c_i^{p,q} \boldsymbol{\theta}_{(i)}^{p,L}, \tag{27}$$

$$\boldsymbol{\omega}_{(0)}^p \leftarrow \sum_{i \in \mathcal{M}} c_i^{p,q} \widehat{\boldsymbol{\omega}}_{(i)}^{p,L} + \sum_{i \notin \mathcal{M}} c_i^{p,q} \boldsymbol{\omega}_{(i)}^{p,L}. \tag{28}$$

The defense mechanism is outlined in Algorithm 6. This protocol can be integrated into Algorithm 3 and 2 as the aggregation algorithm $\mathcal{A}^{agg}$.

---

**Algorithm 6** FRL Defense Aggregation

---

1: **Input:** Submitted local actors $\{\boldsymbol{\theta}_{(i)}^{p,L}\}_{i \leq n}$; Submitted local critics $\{\boldsymbol{\omega}_{(i)}^{p,L}\}_{i \leq n}$.
2: **Output:** Aggregated actor and critic $\boldsymbol{\theta}_{(0)}^p$, $\boldsymbol{\omega}_{(0)}^p$.
3: **for** $i = 1$ **to** $n$ **do**
4:     Server obtains $O_{(i)}^{p,test}$ by $\boldsymbol{\theta}_{(i)}^{p,L}$,
5:     Gets mean reward $\bar{\mathrm{r}}_{(i)}^{p,test}$ from $O_{(i)}^{p,test}$.
6: **end for**
7: **for** $i = 1$ **to** $n$ **do**
8:     Server normalizes the credit $c_i^{p,q} \leftarrow \frac{\bar{\mathrm{r}}_i^{p,q}}{\sum_i \bar{\mathrm{r}}_i^{p,q}}$
9: **end for**
10: Server obtains $\boldsymbol{\theta}_{(0)}^p$ and $\boldsymbol{\omega}_{(0)}^p$ by Eq. (27) and (28).

---

# D  Experiments

## D.1  Additional Settings

### D.1.1  Settings for Main Experiments

**Additional general settings**. We measure the attack cost by the $\ell_2$ distance between the poisoned reward and the ground-truth observed reward. During each local training step, the maximum number of steps before termination is 300 unless restricted by the environment. The learning rate is set to 0.001, and the discount parameter is set to $\gamma = 0.99$.

**Additional settings for robust FRL**. For each model uploaded to the server, we ran it for 10 episodes and collected the mean reward per episode. We then used the normalized rewards as the weights of aggregation. We structure the system with only 2 agents for all environments to ensure that the suspicious agent possesses the maximum possible power to poison the system since a smaller number of agents corresponds to a more potent attack.

### D.1.2  Settings for Additional Experiments

**General settings**. Since VPG is more challenging to attack (refer to Section 6.5.1), in Section D.2 our focus is on attacking VPG-based FRL.

**Targeted poisoning**. For simplicity, we choose the target policy to be a single action, whether in discrete or continuous space. We ensure the chosen environments encompass both discrete and continuous action spaces, featuring diverse cardinalities and dimensions for the action space. Concretely, we set the target policies to be $0 \in \{0, 1\}$ for *CartPole*, $0 \in \{0, 1, 2, 3\}$ for *Lunar Lander*, $3 \in [-3, 3]$ for *Inverted Pendulum*, and $\mathbf{0}^6 \in [-1, 1]^6$ for *Half Cheetah*. We choose environments of different action space: CartPole (two discrete actions), LunarLander (four discrete actions), InvertedPendulum (one continuous dimension), HalfCheetah (six continuous dimensions).

**Multi-agent attack**. We opt for the *CartPole* environment, a relatively simple task, to balance between manageable computation costs and a clear understanding of how a fixed proportion operates in a multi-agent attack scenario. In Section 6.5.1, we have determined that the maximum size of the system one agent can effectively attack is 4 for the CartPole environment and a VPG-based FRL system. Therefore, we initiate the system size from 4, and accordingly, the number of poisoned agents begins at 1. Subsequently, we incrementally enlarge the system size and the corresponding number of poisoned agents while maintaining a constant proportion between the number of poisoned agents and the system size.

## D.2  Results of Experiments

All additional results are obtained from experiments following settings in Section 6.4 and D.1.2.

**Large-budget Attack**. We evaluated the performance of our poisoning with a large budget in Figure (4). The results show that with a larger budget, our method is able to attack much larger systems compared with the constraints that appeared in the case of a small budget, i.e., we can attack up to 100 agents.

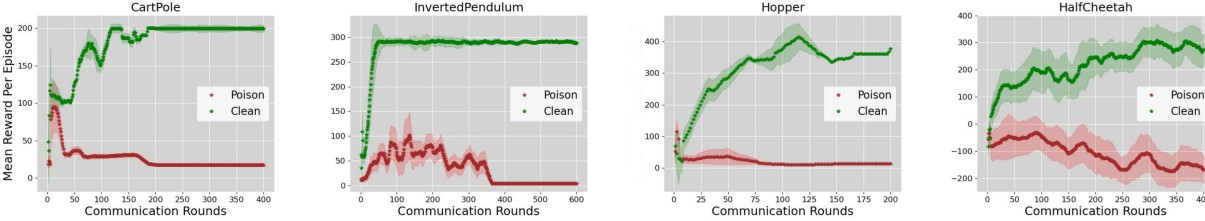

Figure 4: **High-Budget Poisoning**. The red dashed line, labeled as *Poison*, depicts the performance of our adversarial attack in a standard FRL system. The green dashed line, labeled as *Clean*, represents the standard FRL system's performance.

**Multi-agent poisoning**. We have shown that the proportion of malicious agents required to poison a system is consistent regardless of the size of the system. We found that when we increase the system size from 4 agents to 100 agents, a fixed proportion of attackers can always poison the system successfully. This result is depicted in Figure (5).

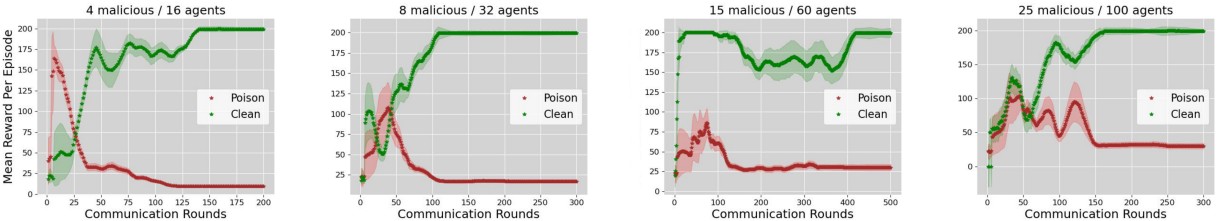

Figure 5: **Multi-Agent Poisoning**. The annotation is the same as Fig. 4.

**Targeted attack.** Our poisoning works well for targeted attack, shown in Figure (6).

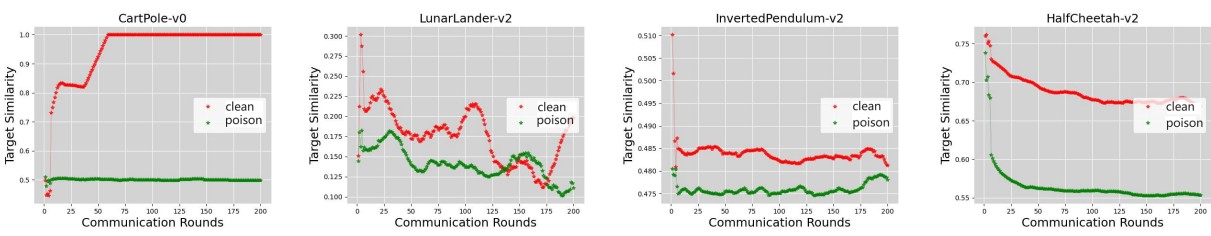

Figure 6: **Targeted attack**. The green dashed line, labeled as *poison*, depicts the performance of our adversarial attack in a standard FRL system. The red dashed line, labeled as *clean*, represents the standard FRL system's performance.

# E  Additional Background

## E.1  Differences between distributed RL and FRL

Distributed RL and FRL are both advanced paradigms in the field of machine learning, but they differ significantly in their architecture and application. Distributed RL involves distributing the computation of a single RL algorithm across multiple machines to accelerate learning and handle larger datasets. This approach focuses on parallelizing the training process to improve efficiency and scalability. On the other hand, FRL is designed to enable multiple independent agents to collaboratively train an RL model without sharing their local data. In FRL, each agent trains a local model using its own data and periodically shares model updates (rather than raw data) with a central server, which aggregates these updates to improve the global model. This method preserves data privacy and is particularly useful in scenarios where data cannot be centralized due to privacy concerns or regulatory restrictions. Thus, while distributed RL aims at enhancing computational efficiency, FRL emphasizes data privacy and decentralized training, making it suitable for applications where data sharing is limited.

## E.2  Different RL algorithms

Reinforcement learning encompasses a variety of algorithms designed to enable agents to learn optimal behaviors through interactions with their environment. Among these, actor-critic algorithms, such as Advantage Actor-Critic (A2C), Asynchronous Advantage Actor-Critic (A3C), and more advanced methods like Proximal Policy Optimization (PPO) and Trust Region Policy Optimization (TRPO), are notable for their efficiency in handling high-dimensional action spaces. PPO, recognized for its robustness and simplicity, prevents large updates that can destabilize training, balancing exploration and exploitation effectively. We chose to focus on PPO and VPG due to their practical balance between performance and computational efficiency. While more advanced algorithms like Deep Deterministic Policy Gradient (DDPG) or Soft Actor-Critic (SAC) offer better performance in certain scenarios, they come with increased complexity and computational costs. We focus on VPG and PPO to ensure the approach remains accessible and practical while achieving competitive performance.

