# OpenReview forum: "Reward Poisoning on Federated Reinforcement Learning"
_TMLR — Accepted by TMLR_

### Review · Reviewer_tQxM · 2024-04-07

**Summary Of Contributions:**

The paper proposes a new training-time attack for federated reinforcement learning. The starts by introducing applications of reinforcement learning, as well as federated reinforcement learning (FRL) before discussing related work on poisoning for different reinforcement learning contexts. Next, the paper outlines contributions, which mostly centers on a new poisoning protocol for FRL, a theoretical analysis of that protocol and experimental evaluation of the protocol.

In Section 2, the paper introduces preliminaries and notations followed by the problem formulation in Section 3. In Section 4, the paper introduces the proposed poisoning protocols methods, including reward poisoning for actor-critic based FRL and policy gradient based FRL with through algorithmic details. Section 5 provides a theoretical analysis of the proposed method and Section 6 provides empirical results across multiple RL environments. Many of the empirical results show that the poisoning methods generally decreases performance of the FRL agent.

**Audience:**

Yes

**Broader Impact Concerns:**

The paper would be strengthened by including a broader impact section that relates to potential misuse of the poisoning method and how to mitigate it.

**Claims And Evidence:**

No

**Requested Changes:**

I request addressing the weakness identified above, including:

* A more thorough discussion of the experimental results, especially when the attack appears to make little difference when considering the variance of the results.
* Adding more relevant results and description into the main paper, such as those found in Appendix C and Appendix D.
* Provide a discussion of different RL algorithms, especially actor-critic algorithms, and why the authors chose to focus on PPO and VGP instead of more advanced algorithms.
* A more thorough discussion on the differences between distributed RL and FRL
* A discussion on why policy gradient does not have local gradient poisoning
* An expansion of the introduction including the application of RL to different design problems, such as design of materials [1][2][3], circuit design [4] and optimization [5] and how the poisoning aspects relate to those cases as well as the cases already identified (e.g., IoT).

[1] Govindarajan, Prashant, et al. "Learning Conditional Policies for Crystal Design Using Offline Reinforcement Learning." Digital Discovery (2024).

[2] Zhou, Zhenpeng, et al. "Optimization of molecules via deep reinforcement learning." Scientific reports 9.1 (2019): 10752.

[3] Ghugare, Raj, et al. "Searching for High-Value Molecules Using Reinforcement Learning and Transformers." arXiv preprint arXiv:2310.02902 (2023).

[4] Roy, Rajarshi, et al. "PrefixRL: Optimization of parallel prefix circuits using deep reinforcement learning." 2021 58th ACM/IEEE Design Automation Conference (DAC). IEEE, 2021.

[5] Chen, Tianlong, et al. "Learning to optimize: A primer and a benchmark." Journal of Machine Learning Research 23.189 (2022): 1-59.

**Strengths And Weaknesses:**

**Strengths:**
* The paper proposes a new method for poisoning FRL agents that can be relevant to enhancing the robustness of FRL.
* The paper provides through details on the algorithmic descriptions.
* The paper provides a useful review of poisoning methods in related RL scenarios.

**Weaknesses:**
* The paper dedicates a lot of space to describing preliminaries and some background that can distract from the main message of the paper. I recommend moving some of the extraneous description into the Appendix and moving relevant experimental results and descriptions from Appendix C and Appendix D into the main paper.
* Some of the empirical results don’t fully support the authors’ claims (e.g., Figure 1 – walker) and the paper currently offers no explanation for those.
* There is no discussion of limitations, future work and/or potential poisoning mitigation strategies.
* Discussion of related work and methods could be strengthened.

---

> ### Author Response · Authors · 2024-07-25
> **Response to Reviewer tQxM**
>
> We appreciate the valuable feedback and the opportunity to improve our work. Below, we address each requested change:
>
> 1. **More thorough discussion of experimental results, especially when the attack appears to make little difference.**
>
> We have updated our experimental results, as shown in Figure 1 in the current revised paper, to address the issue of the unclear gap between our attack and the baseline. In the latest results, we extended the federated rounds from 200 to 400. While the gap was in some cases during the first 200 rounds, with the increased number of federated rounds, our attack can induce a significant decrease in the mean episode reward compared to the baselines.
>
> 2. **Adding more relevant results and description into the main paper.**
>
> Thank you. We will make sure to incorporate key results from Appendix into the main paper in the final revised paper.
>
> 3. **Discussion of different RL algorithms, especially actor-critic algorithms.**
>
> Reinforcement Learning (RL) encompasses a variety of algorithms designed to enable agents to learn optimal behaviors through interactions with their environment. Among these, actor-critic algorithms, such as Advantage Actor-Critic (A2C), Asynchronous Advantage Actor-Critic (A3C), and more advanced methods like Proximal Policy Optimization (PPO) and Trust Region Policy Optimization (TRPO), are notable for their efficiency in handling high-dimensional action spaces. PPO, recognized for its robustness and simplicity, prevents large updates that can destabilize training, balancing exploration and exploitation effectively. We chose to focus on PPO and VPG due to their practical balance between performance and computational efficiency. While more advanced algorithms like Deep Deterministic Policy Gradient (DDPG) or Soft Actor-Critic (SAC) offer better performance in certain scenarios, they come with increased complexity and computational costs. We focus on VPG and PPO to ensure the approach remains accessible and practical while achieving competitive performance.
>
> We will add such discussion into the experiment setting section in our revised version.
>
> 4. **Discussion on the differences between distributed RL and FRL**
>
> Distributed Reinforcement Learning and FRL are both advanced paradigms in the field of machine learning, but they differ significantly in their architecture and application. Distributed RL involves distributing the computation of a single RL algorithm across multiple machines to accelerate learning and handle larger datasets. This approach focuses on parallelizing the training process to improve efficiency and scalability. On the other hand, FRL is designed to enable multiple independent agents to collaboratively train an RL model without sharing their local data. In FRL, each agent trains a local model using its own data and periodically shares model updates (rather than raw data) with a central server, which aggregates these updates to improve the global model. This method preserves data privacy and is particularly useful in scenarios where data cannot be centralized due to privacy concerns or regulatory restrictions. Thus, while distributed RL aims at enhancing computational efficiency, FRL emphasizes data privacy and decentralized training, making it suitable for applications where data sharing is limited.
>
> We will add such discussion in our final revised paper. Besides, we have included a thorough discussion on distributed RL and federated RL from the view of poisoning in Section 1.1, which are the two paragraphs starting with “Poisoning in Distributed RL” and “Poisoning in FRL”, respectively.
>
> 5. **Explanation of why policy gradient does not have local gradient poisoning**
>
> If we understand the reviewer correctly, the question is, "In policy-gradient based FRL, why is the local gradient not poisoned by our poisoning method?" The answer is that the local gradient is indeed poisoned by our method. In fact, not only the gradient but everything related to rewards (i.e., objective, model parameters) is poisoned through our reward poisoning. We will add several lines to clarify this issue in the revised manuscript.
>
> 6. **Expansion of the introduction to include RL applications in design problems**
>
> The required literature is cited as applications of RL (highlighted in blue in the first paragraph in Introduction). These domains are also vulnerable to training-time poisoning, as RL heavily relies on data collected from the environment. For instance, in material design, RL is susceptible to poisoned material data, while in circuit design, it can be compromised by misleading prefix graphs and actions. Related poisoning aspects are added (highlighted in brown in the third paragraph in Introduction)
>
> 7. **Broader Impact Concern**
>
> Thank you. In the revised manuscript, we will include a thorough discussion to elaborate on the broader impacts of our work.
>
> Thank you again for your valuable feedback, and we hope our new changes will be found satisfactory.

---

### Review · Reviewer_BvyU · 2024-06-03

**Summary Of Contributions:**

The work looks into a new area of problem in poisoning attacks for federated reinforcement learning. This framework is designed for policy-based FRL and can be applied to both policy-gradient local RL and actor-critic local RL scenarios.

The authors provided theoretical analysis and conducted experiments to evaluate the poisoning attack method.

**Audience:**

Yes

**Broader Impact Concerns:**

This is not provided. Since the work focuses on the security of AI, I suggest adding a section on this.

**Claims And Evidence:**

No

**Requested Changes:**

- Clarify the challenges of the problem and the contribution of the work
- Add the RL poisoning work as baselines and compare the performance (see above)
- clarify the evaluation section in terms of the number of agents used, and evaluate how the number of agent and random selection of agents affect the performance of the proposed method
- clarify the threat model
- improve writing (see above)

**Strengths And Weaknesses:**

## Strength
- This work is looking into a new problem of poisoning attacks for FRL
- Theoretical analysis is provided to guarantee the attack method can decrease the global objective.

## Weakness
- The contribution seems limited. It is unclear what the challenges are for applying poisoning attacks to FRL compared to poisoning attacks for RL.
- For baselines in the evaluation section, why are the many existing cited RL poisoning methods not compared in this work, you can apply those methods for a single agent in the RL setting. It is hard to see the performance advantage of your proposed method when not compared to those baselines.
- In the evaluation section, it is unclear how the method performs as you have different numbers of agents. The number of agents for each case is also not clearly stated (only mentioned 3 to 4 agents for the VPG-base FRL case). The evaluation results for poison VPG-base FRL seem to have similar effects as the random attack.
- The attack method involves a public critic and a private critic which seems to require intervention in the training processes and weight upload, so it's not only just modifying the rewards. Is that a realistic threat model?
- I suggest improving the writing of the formulation section to make it simpler for readers to understand your work. For instance, 1) you define O as the sequence of state, action, and reward in section 2, and refer to them as observations in section 3.3. This can be easily confused with the "observation" in partial observable RL problems. 2)when you use the notation of i<n (equation 2) and i!=n (table 1 & 2), is there a reason why they have to be different? if not, it's better to keep them consistent.

---

> ### Author Response · Authors · 2024-07-25
> **Response to Reviewer BvyU**
>
> We appreciate the detailed feedback and suggestions for improving our manuscript. Below, we address each of the reviewer's requests:
>
> 1. **Clarification of the Challenges and Contribution**
>
> * Challenges: Applying existing RL poisoning techniques to FRL presents challenges. Many prior work, such as those by Zhang (2020) and Rakhsha (2020), rely on unrealistic assumptions, including the attacker having complete knowledge of the environment MDP, which is often impractical. Additionally, some approaches, like TrojDRL (Panagiota, 2020), target RL agents' actions rather than rewards, making them incompatible with our framework. Furthermore, the effectiveness of certain mechanisms, such as VA2C-P (Sun 2020), diminishes in federated settings due to small local training steps, which lead to frequent reinitialization of the adversarial critic and impaired learning of the poisoned actor’s value function. Nevertheless, we have included the federated version of VA2C-P as a baseline in our revised paper (Figures 1 and 2, yellow line), and our approach significantly outperforms this baseline.
>
> * Contribution: We developed a method that does not require access to the MDP, addressing the knowledge limitations presented in prior RL approaches. Additionally, our approach is tailored specifically for federated contexts, particularly for actor-critic-based FRL. By using a private critic which leverages historical local training rounds, our method effectively overcomes the diminished effectiveness of prior RL poisoning techniques when directly applied to federated frameworks.
>
> In our revised manuscript, challenges are discussed in the 5th paragraph in Introduction highlighted blue, and contribution is updated at the first bullet point of the contribution part located at the end of Introduction (highlighted blue).
>
> 2. **Inclusion of RL Poisoning Work as Baselines**
>
> We have included the federated extension of VA2C-P (a prior RL poisoning) as one of our baselines. This choice was made because VA2C-P is compatible with our reward-manipulating scheme and adheres to the knowledge limitation that the attacker lacks access to the environment's MDP. In contrast, other prior RL poisoning methods either present compatibility issues or violate this knowledge limitation. The updated results (Figures 1 and 2) demonstrate that our approach significantly outperforms this additional baseline.
>
> 3. **Clarification of the Evaluation Section**
>
> * Criteria for the Number of Agents: For each environment, we present the **maximum** size of a federated system that our poisoning method can effectively target. For instance, 4 for CartPole indicates that our approach can successfully poison systems with 1, 2, 3, or up to 4 agents, but would not be effective for a 5-agent system. This criterion is highlighted in blue in the captions of Figures 1 and 2.
>
> * Effects of Poisoning VPG-based FRL Compared to Random Attack: We have updated our experimental results (Figure 1 of the revised paper) to clarify the differences between our attack and the random attack baseline. In the latest results, we extended the federated rounds from 200 to 400. Although the gap between our attack and the random attack baseline was noticeable in some cases during the initial 200 rounds, the extended number of federated rounds reveals a significant decrease in the mean episode reward due to our attack compared to the baselines.
>
> 4. **Clarification of the Threat Model**
>
> We assert that our attack is realistic and that reward manipulation is the only method we use for poisoning, without intervening in the training process of victim. Our poisoning remains practical whether agent n is a victim or an attacker:
>
> * If agent n is a victim, it possesses both an actor and a critic model. The critic model used for communication with the server is called public, while the private critic is retained by the attacker, but not agent n. Thus, the victim is poisoned solely through the reception of manipulated rewards.
>
> * If agent n is a malicious agent, it has full control over the actor model and both critic models, thus such poisoning model is practical. Nonetheless, the poisoning of models communicated to the server is executed through reward manipulation.
>
> Therefore, regardless of whether agent n is a victim or an attacker, our approach remains both realistic and practical. We will include this clarification in the revised version.
>
> 5. **Improvement of Writing**
> We will rename the Gathered Sequence of (state, action, and reward) from O to G to avoid confusion and make corresponding adjustments in the future version. Additionally, we have updated i \neq n to i<n (highlighted in blue) in Table 1 and 2 to ensure consistency with Equation 2.
>
> 6. **Broader Impact Concern**
>
> Thank you. In the revised manuscript, we will include a thorough discussion to elaborate on the broader impacts of our work.
>
> Thank you again for your valuable feedback, and we hope our new changes will be found satisfactory.

---

### Review · Reviewer_jXvH · 2024-07-13

**Summary Of Contributions:**

This submission introduced a novel reward poisoning mechanism for FRL under both general policy-based and actor-critic algorithms. The authors theoretically demonstrate the importance of poisoning attack w.r.t. the risks during real-world scenarios, and highlight the problem that draws the attention of the community. By conducting a range of experiments on various OpenGYM environments using popular RL models, the authors show that the proposed protocol is effective in poisoning FRL systems and inspires future research to design more robust algorithms to protect FRL against poisoning attacks.

**Audience:**

Yes

**Claims And Evidence:**

No

**Requested Changes:**

See the weakness, which should be taken into consideration to improve the submission. In detail,

(1) the writing on some critical claims should be improved to make them more proper and are well supported with some facts or discussion.

(2) The experiments can be enhanced by considering some varying factors in FRL.

**Strengths And Weaknesses:**

Strengths:

(1) The authors study an interesting and important problem, i.e., the poisoning attack in FRL, which should be paid more attention in the real-world scenarios given the vulnerability of FL and RL as well as their composition. Specifically the authors are among the first attempts to show the high risk of the potential poisoning to FRL.

(2) The authors design a novel poisoning attack, which is generally effective under different FRL models, such as actor-critic-based FRL and policy-based FRL. Besides, the authors provide a principled proof to show the effect of the proposed reward poisoning in sabotaging the system, specifically by degrading the global objective.

(3) Through extensive experiments under the OpenGYM environments with varying difficulty levels, the authors shows the consistent effectiveness by comparing with various baseline models and assessments of different (targeted and nontargeted) poisoning types in the mainstream RL algorithms like VPG and PPO.

Weakness:

Despite promise, there are still some concerns that should be carefully addressed, which I summarized as follows.

(1) Some claims in the manuscript is not very proper. For example, the first sentence in the abstract "Federated learning (FL) has become a popular tool for solving traditional Reinforcement Learning (RL) tasks." FL is not a popular tool to RL when there is no constrain on the data governess. Therefore, the authors should point out the application range of FL regarding RL. The beginning of the third paragraph in the introduction "Poisoning in FRL is practical and harmful. The inherent nature of FL and RL amplifies the susceptibility to poisoning (training-time attacks) when combined into FRL." Some evidence can be given to show the potential amplifying effect w.r.t. the poisoning. Or you can give more intuitive explanation to convince the readers about your point.

(2)  The authors can further explain the rationality about Theorem 6, as the even the global objective can be smaller under some poisoning budget, the optimization eventually decides the convergence of the model. It will be more helpful to enrich the discussion about the theorem so that both the implication and limitation are covered.

(3) The experiments can be enhanced, as many factors like the ranging number of the participants in FRL (not the poisoning agents), the different configurations of one environment to show the consistency etc, has not comprehensively considered.

---

> ### Author Response · Authors · 2024-07-25
> **Response to Reviewer jXvH**
>
> We appreciate your detailed feedback and the opportunity to address the concerns raised. Below, we respond to each point and outline the changes we will make:
>
> 1. **Address Application Range of FL in RL**
>
> We have revised the first sentence in the abstract to: “Federated learning (FL) has become a popular tool for solving traditional Reinforcement Learning (RL) tasks, particularly when individual agents need to collaborate due to low sample efficiency but are concerned about data privacy.” This change is highlighted in blue.
>
>
> 2. **Give Intuitive explanation and evidence on potential amplifying effect of FRL w.r.t. the poisoning.**
>
> At the local RL level, RL agents are dependent on feedback from their environment, making the system susceptible to training-time data corruption (Zhang et al., 2020; Banihashem et al., 2022). For instance, a chatbot might be misled by a small group of Twitter users, or recommendation systems could be manipulated by fake clicks.
>
> At the Federated Learning (FL) framework level, the privacy-protecting nature of FL leads to a lack of transparency in individual local training, rendering the system vulnerable to training-time attacks from malicious agents (Fang et al., 2020; Bagdasaryan et al., 2020; Bhagoji et al., 2019; So et al., 2020).
>
> Consequently, since both the central FL protocol and the local RL training mechanism are susceptible to poisoning, attacks on Federated Reinforcement Learning (FRL) are inherently easier to implement and more damaging.
>
> We have presented the examples and intuitive explanation above in the third paragraph of the Introduction, highlighted in blue.
>
>
> 3. **Clarification on Theorem 6:**
>
> Theorem 6 does not indicate convergence induced by poisoning. The result we obtain is $\hat{J}\_{(0)}^p \leq J\_{(0)}^p - \alpha$, where the RHS represents an upper bound for the LHS. This means that $\alpha$ represents the **minimum** gap caused by poisoning, and the actual gap could be larger than $\alpha$. Therefore, the theorem shows that $\alpha < gap < \infty$, which does not indicate convergence, but provides a guarantee on the minimum gap induced by poisoning.
>
> We provide a detailed discussion in the paragraph following Theorem 6, highlighted in blue, where we explore the relationship between the guaranteed gap and factors such as the attack budget and learning rate.
>
>
> 4. **Clarify Consistency of experiments by considering some varying factors in FRL.**
>
> We ensure the consistency of our experiments by evaluating various configurations within a single environment, including:
>
> a) *Different Number of Participants*: For each environment, we conducted experiments with varying numbers of participants. In Figures 1 and 2, the captions indicate the maximum size of a federated system that our poisoning method can effectively target. For example, a value of 4 for CartPole signifies that our approach can successfully poison systems with 1, 2, 3, or up to 4 agents, but it would not be effective for a 5-agent system.
>
> b) *Different Number of Attackers*: We performed experiments with various numbers of attackers. The results are shown in Figure 5.
>
> c) *Different Attack Budgets*: We investigated both small and large poisoning budgets. Results for small budgets are presented in Figures 1, 2, and 3, while results for large budgets are shown in Figure 4.
>
> d) *Different Poisoning Purposes*: We conducted experiments with both targeted and non-targeted poisoning. Results for non-targeted poisoning are presented in Figures 1, 2, 3, 4, and 5, whereas results for targeted poisoning are shown in Figure 6.
>
>
> 5. **Broader Impact Concern**
>
> Thank you. In the revised manuscript, we will include a thorough discussion to elaborate on the broader impacts of our work.
>
> Thank you again for your valuable feedback, and we hope our new changes will be found satisfactory.

---

> > ### Comment · Reviewer_jXvH · 2024-08-12
> >
> > Thank you for the thorough response and revision. Now, my concerns have been well addressed. I recommend towards acceptance.

---

### Decision · Action_Editor_9sZ9 · 2024-09-04

**Recommendation:** Accept with minor revision

**Comment:**

This paper ntroduced a novel reward poisoning mechanism for FRL under both general policy-based and actor-critic algorithms. The authors theoretically demonstrate the importance of poisoning attack w.r.t. the risks during real-world scenarios, and highlight the problem that draws the attention of the community. By conducting a range of experiments on various OpenGYM environments using popular RL models, the authors show that the proposed protocol is effective in poisoning FRL systems and inspires future research to design more robust algorithms to protect FRL against poisoning attacks.

The authors study an interesting and important problem, i.e., the poisoning attack in FRL, which should be paid more attention in the real-world scenarios given the vulnerability of FL and RL as well as their composition.The authors design a novel poisoning attack, which is generally effective under different FRL models, such as actor-critic-based FRL and policy-based FRL. Through extensive experiments under the OpenGYM environments with varying difficulty levels, the authors shows the consistent effectiveness by comparing with various baseline models and assessments of different (targeted and nontargeted) poisoning types in the mainstream RL algorithms like VPG and PPO. However, the contribution seems a bit limited. It is unclear what the challenges are for applying poisoning attacks to FRL compared to poisoning attacks for RL. Therefore, it is better to clarify the challenges of the problem and the contribution of the work, and add the RL poisoning work as baselines and compare the performance. Based on three qualified reviews, this paper can be accepted with minor revision and the authors are encouraged to merge the comments into their update versions.

**Audience:**

Yes

**Claims And Evidence:**

Yes

---

> ### Author Response · Authors · 2024-09-23
> **Response to Action Editor 9sZ9**
>
> We would like to express our sincere gratitude for the valuable feedback and recommendations.
>
> Below, we address the required revisions:
>
> 1. **Clarification on Challenges and Contribution, especially regarding Poisoning FRL Compared to Poisoning RL**
>
>     We clarify the challenges of applying prior RL poisoning to the federated setting and our corresponding contribution as follows:
>
>     * **Challenges**: Applying existing RL poisoning techniques to FRL poisoning presents challenges.
>
>         a) Many prior works, such as those by Zhang et al., 2020 [1] and Rakhsha et al., 2020 [2], rely on unrealistic assumptions, including the attacker having complete knowledge of the environment MDP, which is often impractical.
>
>         b) Additionally, some approaches, like TrojDRL (Panagiota et al., 2020 [3]), target RL agents' actions rather than rewards, making them incompatible with our framework.
>
>         c) Furthermore, the effectiveness of certain mechanisms, such as VA2C-P (Sun et al., 2020 [4]), diminishes in federated settings due to small local training steps, which lead to frequent reinitialization of the adversarial critic and impaired learning of the poisoned actor’s value function.
>
>     * **Contribution**: Our method addresses the challenges of prior RL poisoning in two key aspects: (a) *Adherence to federated privacy code.* Our method strictly avoids accessing the environment’s MDP, a code often violated by previous RL poisoning. (b) *Leveraging historical federated rounds.* Our double-critic mechanism leverages historical data from prior federated rounds, overcoming the reduced poisoning effectiveness of prior RL attacks applied to federated contexts.
>
>     In our revised manuscript, challenges are updated in Section 1.1 highlighted blue, and contribution is updated in Section 1.3 highlighted blue.
>
>     [1] *Xuezhou Zhang, Yuzhe Ma, Adish Singla, and Xiaojin Zhu. Adaptive reward-poisoning attacks against reinforcement learning. In International Conference on Machine Learning, pp. 11225–11234. PMLR, 2020.*
>
>     [2] *Amin Rakhsha, Goran Radanovic, Rati Devidze, Xiaojin Zhu, and Adish Singla. Policy teaching via environment poisoning: Training-time adversarial attacks against reinforcement learning. In International Conference on Machine Learning, pp. 7974–7984. PMLR, 2020.*
>
>     [3] *Kiourti Panagiota, Wardega Kacper, Susmit Jha, and Li Wenchao. Trojdrl: Trojan attacks on deep reinforcement learning agents. in proc. 57th acm/ieee design automation conference (dac), 2020, march 2020. In Proc. 57th ACM/IEEE Design Automation Conference (DAC), 2020, 2020.*
>
>     [4] *Yanchao Sun, Da Huo, and Furong Huang. Vulnerability-aware poisoning mechanism for online rl with unknown dynamics. arXiv preprint arXiv:2009.00774, 2020.*
>
>
> 2. **Comparison with RL Poisoning Baselines**
>
>     In our revised manuscript, we have included the federated extension of VA2C-P [4] as one of our baselines. This choice was made because VA2C-P is compatible with our reward-manipulating scheme and adheres to the knowledge limitation that the attacker lacks access to the environment's MDP. In contrast, other prior RL poisoning methods either present compatibility issues or violate this limitation. The updated results in Figures 1 and 2 demonstrate that our approach significantly outperforms this additional baseline.
>
>     The justification for choosing VA2C-P as an RL poisoning baseline is provided in Section 6.3, bullet point 1, highlighted in blue. The experimental results for the RL poisoning baseline are presented in Figures 1 and 2 as the yellow lines. Additionally, the discussion of these results has been updated in Section 6.5.2, also highlighted in blue.
>
> 3. **Further Discussion**
>
>     In our revised manuscript, we have expanded the discussion in response to the reviewers’ comments:
>
>     * A discussion on the broader impacts of our work has been added in Section 7, highlighted in blue.
>
>     * A discussion on the differences between distributed RL and FRL is included in Appendix E.1, also highlighted in blue.
>
>     * A discussion of different RL algorithms is presented in Appendix E.2, highlighted in blue.
>
> Once again, we appreciate the constructive comments and hope that our additional changes will be found satisfactory.